

**1  Dimethylsulfoniopropionate (DMSP) and dimethylsulfide (DMS) cycling**

**2  across contrasting biological hotspots of the New Zealand Subtropical**

**3  Front**

Martine Lizotte[1], Maurice Levasseur[1], Cliff S. Law[2#], Carolyn F. Walker[2], Karl A. Safi[3],
Andrew Marriner[2], Ronald P. Kiene[4]
Corresponding author: martine.lizotte@qo.ulaval.ca
Tel.: (418) 656-2131 #6274
Fax.: (418) 656-2339
Submitted to a Special Issue of ACP – OS on SOAP
Running Title: Hotspot DMSP and DMS cycling in the NZ Subtropical Front
Key words: Dimethylsulfoniopropionate (DMSP) – Dimethylsulfide (DMS) – Bacteria –
Sulfur cycling – New Zealand – Chatham Rise – Phytoplankton bloom – Subtropical
Front (STF) – Subtropical Convergence
[1] Université Laval, Department of biology (Québec-Océan), Québec City, Québec,
Canada.
[2] National Institute of Water and Atmospheric Research, Wellington, New Zealand
[#] University of Otago, Department of Chemistry, Dunedin, New Zealand
[3] National Institute of Water and Atmospheric Research, Hamilton, New Zealand
[4] University of South Alabama, Department of Marine Sciences, Mobile, USA


## 1 Abstract

The oceanic frontal region above the Chatham Rise east of New Zealand was investigated during the late austral summer season in February and March 2012. Despite its potential importance as a source of marine-originating and climate-relevant compounds, such as dimethylsulfide (DMS) and its algal precursor dimethylsulfoniopropionate (DMSP), little is known of the processes fuelling the reservoirs of these sulfur (S) compounds in the water masses bordering the Subtropical Front (STF). This study focused on the two opposing fates of DMSP-S following its uptake by microbial organisms: either its conversion into DMS, or its assimilation into bacterial biomass. Sampling took place in three phytoplankton blooms (B1, B2 and B3) with B1 and B3 occurring in relatively nitrate-rich, dinoflagellate-dominated Subantarctic waters, and B2 occurring in nitrate-poor Subtropical waters dominated by coccolithophores. Concentrations of total DMSP ($DMSP_t$) and DMS were high across the region, up to 160 nmol $L^{-1}$ and 14.5 nmol $L^{-1}$, respectively. Pools of $DMSP_t$ measured in this study showed a strong association with overall phytoplankton biomass proxied by chlorophyll $a$ ($r_s = 0.83$) likely because of the persistent dominance of dinoflagellates and coccolithophores, both DMSP-rich taxa. Heterotrophic microbes displayed low S assimilation from DMSP (less than 5%) likely because their S requirements were fulfilled by high DMSP availability. Rates of bacterial protein synthesis were significantly correlated with concentrations of dissolved DMSP ($DMSP_d$, $r_s = 0.86$) as well as with the microbial conversion efficiency of $DMSP_d$ into DMS (DMS yield, $r_s = 0.84$). Estimates of the potential contribution of microbially-mediated rates of DMS production (0.1 - 27 nmol $L^{-1}$ $d^{-1}$) to the near-surface concentrations of DMS suggest that bacteria alone could not have sustained DMS pools at most stations, indicating an important role for phytoplankton-mediated DMS production. The findings from this study provide crucial information on the distribution and cycling of DMS and DMSP in a critically under-sampled area of the global ocean, and they highlight the importance of oceanic fronts as hotspots of the production of marine biogenic S compounds and as potential sources of aerosols particularly in regions of low anthropogenic perturbations such as the frontal waters of the Southern Hemisphere.



## 2 Introduction


In oceanic waters, the gas dimethylsulfide (DMS) is the predominant biogenic compound
contributing to the flux of sulfur (S) from the hydrosphere to the atmosphere (Bates et al.,
1992; Simó, 2001) with 17.6 to 34.4 Tg of S estimated to be transferred annually (Lana et
al., 2011). DMS has gained notoriety over several decades of research on the grounds of
its potential role linking ocean biology and the climate (Andreae et al., 1985; Charlson et
al., 1987; Lovelock et al., 1972). Produced through the enzymatic cleavage of its marine
algae-derived precursor, dimethylsulfoniopropionate (DMSP), DMS ventilates to the
marine atmospheric boundary layer (Liss et al., 1997) where it is oxidized, mainly by the
hydroxyl radical OH (Andreae and Crutzen, 1997). DMS oxidation products may
influence the atmospheric radiative budget via their role in aerosol properties and cloud
condensation as well as their contribution to a persistent stratospheric aerosol layer, or
Junge layer (Gondwe et al., 2003; Marandino et al., 2013). The significance of DMS-
derived particles in affecting the Earth's cloudiness and albedo is largely determined by
the relative importance of atmospheric DMS oxidation products compared to other
airborne particles originating from, for example, sea salts, dust and anthropogenic
pollutants (Quinn and Bates, 2011). As such, areas without significant dust or
anthropogenic particle inputs may offer productive grounds for new particle formation
emanating from DMS.

Because DMS is of biogenic origin, factors controlling the distribution and productivity
of marine plankton play a large role in shaping DMS dynamics and standing stocks.
Oceanic frontal and convergence zones are regions of intense mesoscale turbulence
displaying enhanced levels of chlorophyll-*a* (Belkin et al., 2009) detectable from space
(Weeks and Shillington, 1996). The heightened biological activity in these regions (Llido
et al., 2005) is thought to lead to intensified carbon drawdown on seasonal timescales
(Metzl et al., 1999) as well as high concentrations of DMS (Holligan et al., 1987; Matrai
et al., 1996). These productive regions sometimes form unique biogeographic habitats of
their own such as the Subtropical Convergence province proposed by (Longhurst, 2007).
Nearly encircling the entire globe in a meridional band between 35-45$^\circ$S, the Subtropical
Convergence, or hereafter termed the Subtropical Front (STF), spreads for the most part
across remote regions of the planet where anthropogenic sources of atmospheric



compounds exert subordinate influence on local aerosol patterns compared to natural
sources. Modeling-based evidence suggests that cloud condensation nuclei seasonality is
driven mainly by DMS oxidation in this part of the ocean (Gondwe et al., 2003; Kloster
et al., 2006; Vallina et al., 2006). Episodic phytoplankton bloom events in the STF occur
mostly in austral spring-summer, with varying lifetimes of 8 to 60 days (Llido et al.,
2005).  Upon reaching the Islands of New Zealand (NZ), the STF runs North along the
eastern continental shelf break over the Chatham Rise, a relatively shallow (250-350 m)
and productive seamount (Bradford‐Grieve et al., 1997; Sutton, 2001).

While waters over Chatham Rise are recognized as biological hotspots (Rowden et al.,
2005) supporting large phytoplankton blooms visible from space (Sadeghi et al., 2012),
as well as accumulations of zooplankton and pelagic fish (Tracey et al., 2004), little is
known of their productivity in terms of climate-relevant gases such as DMS. The latest
DMS climatological exercise by Lana et al. (2011) shows that for the New Zealand
Coastal (NEWZ) province only 6 data points are available (together averaging less than
$< 3$ nmol DMS $L^{-1}$), with the temporal extent limited to the month of October. The
biological cycling of DMS in this region thus remains surprisingly under documented and
mainly restricted to the continental shelf of New Zealand's North Island (Walker et al.,
2000). The bordering ocean provinces comprised of the Subantarctic Water Ring (SANT)
and the South Subtropical Convergence (SSTC) have higher data coverage with greater
temporal resolution, displaying monthly averages of ca. 5 nmol DMS $L^{-1}$ (December) and
ca. 10 nmol DMS $L^{-1}$ (January), respectively. These results suggest that greater variation
in DMS concentration might be expected in the NEWZ province, a proposition confirmed
by a recent study showing DMS concentrations in surface waters over Chatham Rise
spanning an order of magnitude (from ca. 4 to 40 nmol $L^{-1}$, see Walker et al., 2016). It is
thus paramount to better constrain the factors that affect DMS concentrations in surface
waters above topographic plateaus and in oceanic convergence zones in view of the
potential for phytoplankton blooms in these biologically active systems.
Phytoplankton bloom dynamics, particularly their speciation and their growth phases,
from onset to senescence, are thought to play major roles in shaping the distribution of
DMS firstly through the variable biosynthesis of DMSP by different members of the
phytoplankton community (Keller, 1989; Matrai and Keller, 1994). DMSP production is



a widespread process in phytoplankton but its magnitude varies substantially among taxa,
from non-detectable among certain cyanobacteria and diatoms, to considerable amounts
(up to 400 mmol DMSP $L^{-1}$ of cell volume) within groups such as dinoflagellates and
prymnesiophytes (Keller, 1989). Furthermore, physicochemical conditions encountered
by algal populations in their environment, such as nutrient repletion or depletion, doses of
solar radiation, oxidative stresses, and modifications in salinity or temperature may also
impact the production of DMSP, as algal cells up- or down-regulate their production to
cope with these external pressures (Simó, 2001; Stefels et al., 2007; Sunda et al., 2002).
DMSP is released into the aqueous environment largely because of cell disruption
following aging, grazing or viral attack (Dacey and Wakeham, 1986; Turner et al., 1988)
and, to a lesser extent, by healthy algae via active exudation (Laroche et al., 1999). Some
non-DMSP producing algal species are thought to take up available dissolved DMSP
directly from the medium and assimilate sulfur from DMSP through a process yet to be
identified (Vila-Costa et al., 2006a).

Beyond its role as the precursor of DMS, DMSP also holds global biogeochemical
significance as a prominent source of reduced S and carbon (C) for marine heterotrophic
microorganisms (Kiene et al., 2000; Simó and Dachs, 2002). Depending on bacterial
requirements for either S or C and the relative contribution of DMSP to the overall
oceanic S pool (Kiene et al. 2000; Levasseur et al 1996; Pinhassi et al. 2005), at least two
very different and competing outcomes are involved from the bacterial catabolism of
DMSP: one producing DMS, the climatic relevant gas, the other producing methanethiol
(MeSH), an important microbial substrate (Kiene and Linn, 2000b). The relative
importance of these competing pathways varies widely in nature and the yield of DMS
from $DMSP_d$ (moles of DMS produced from moles of DMSP consumed) may vary from
2 to 100%. The factors controlling them, however, are still poorly understood (Kiene et
al., 2000; Simó and Pedrós-Alió, 1999). Bacterial production of DMS is not the sole
pathway bolstering reservoirs of DMS in marine waters: certain species of autotrophic
phytoplankton can also directly cleave DMSP into DMS. Although the particular
enzymatic reactions that govern DMSP breakdown are not fully characterized (Todd et
al., 2007), most reactions are attributed to DMSP lyases (Alcolombri et al., 2015; Schafer
et al., 2010; Stefels et al., 2007). What controls the contribution of either process

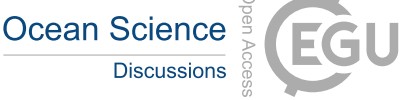

(autotrophic or heterotrophic DMSP to DMS conversion) in fuelling DMS stocks remains
unclear but appears to vary extensively (Lizotte et al., 2012). While there are multiple
sources of DMS, there are also multiple sinks, including bacterial consumption, sunlight
oxidation and finally a small fraction (< 10%) of the produced DMS may ventilate to the
marine boundary layer (Malin, 1997) where its oxidation products, namely sulfate aerosol
particles, can potentially influence the Earth's radiation budget directly through solar
backscattering and indirectly by seeding brighter and longer-lived clouds (Albrecht,
1989; Ångström, 1962; Charlson et al., 1987; Twomey, 1977).

Gaining insight into how marine microorganisms influence the Earth's atmosphere and
climate are topics of prime interest for the international scientific community and at the
core of investigations implemented by the Surface Ocean Aerosol Production (SOAP)
programme (Law et al. this issue). Under the auspices of SOAP, this study specifically
explored two competing bacterial DMSP catabolic processes: (1) DMSP cleavage
(Visscher et al., 1991; Yoch et al., 1997), a non S-assimilating pathway allowing bacteria
to utilize the carbon contained in DMSP in the form of acrylate while the sulfur moiety is
released    as    DMS    (Kiene    et    al.,    2000;    Yoch,    2002);    (2)    DMSP
demethylation/demethiolation (Taylor and Gilchrist, 1991; Taylor and Visscher, 1996), a
S-assimilatory pathway leading to MeSH production, a portion of which is incorporated
directly into methionine, and subsequently into proteins by marine bacteria (Kiene et al.,
1999). The later pathway is thus linked to sulfur assimilation but also yields a methyl
group that can be used as a carbon source (Kiene and Linn, 2000a; Yoch, 2002).
The present study was carried out during austral summer within three autotrophic blooms,
each exhibiting varying phytoplankton assemblages and developmental stages, and
sourced within the upper surface mixed layers of a section of the Subtropical Front over
Chatham Rise east of New Zealand. To our knowledge, the results presented here are the
first rate measurements made in the highly productive ocean region east of New Zealand,
and provide much needed information on the concentrations and cycling of DMS and
DMSP in connection to the "microbial maze" (Malin, 1997) in frontal zones.






**3 Methodological approach**
*3.1 Oceanographic setting*
Large-scale remote sensing through MODIS (Aqua and Terra) and underway
instrumentation for Chl *a*, pCO$_2$, λ660 backscatter, and DMS were employed to detect
biologically productive areas near Mernoo Gap and the eastern end of Chatham Rise (see
Table 1 as well as Bell et al. (2015) and Law et al. (this issue) for further details on
voyage track, location map and biogeochemical characteristics of the sampling area).
Briefly, areas located between 43-45°S east of New Zealand were evaluated for relevant
bloom bio-indicators, and hotspots were marked by a drifting Spar Buoy for further
subsampling. Three distinct blooms were identified and each was followed during
relatively short (<10 days) Lagrangian-type surveys. Nomenclature used by Bell et al.,
(2015) and Law et al. (this issue) to describe these three sampling clusters, i.e. bloom 1,
bloom 2, and bloom 3 (hereafter referred to as clusters B1, B2 and B3) are also used in
this paper to simplify cross-referencing and data comparisons.

Solar radiation dose (SRD in W m$^{-2}$) was calculated using Eq. (1):

$$\text{SRD} = \frac{I_0}{k \cdot \text{MLD}} \cdot \left(1 - e^{-k \cdot \text{MLD}}\right) \qquad (1)$$


where I$_0$ represents the daily-averaged irradiance (in W m$^{-2}$) measured using an Eppley
Precision Spectral Pyronometer (285-2800 nm), k (in m$^{-1}$) are estimates of vertical diffuse
attenuation coefficients based on Photosynthetically Active Radiation (PAR) offset
between two depths (2 m and 10 m), MLD is the mixed layer depth defined as the point at
which a 0.2°C difference from the sea surface temperature occurred and was calculated
according to Kara et al. (2000).
Ambient NO$_3$- concentrations were measured using colorimetric detection by segmented
autoanalyser as described by (Law et al., 2011). Total chlorophyll *a* (Chl *a*: Whatman
glass fibre GF/F filtered) concentrations were determined using 90% acetone extraction
by the fluorometric technique with a Turner Design fluorometer after Strickland and
Parsons (1972). Bacterial samples were frozen in liquid nitrogen (Lebaron et al., 1998)
and thawed immediately before counting by flow cytometry following the methods



described in Safi et al. (2007). Coccolithophore abundance was determined using optical
microscopy as described in Chang and Northcote (2016).

*3.2 Microbial DMSP catabolism incubations*
Surface seawater samples were collected from a rigid-hulled inflatable boat away from
the ship, between 7h00 and 9h00 (NZST) in the morning, with a novel apparatus dubbed
"the sipper". The latter consists of a floating tubing array with peristaltic pump allowing
the sampling of the undisrupted first 1.6 m of the upper mixed layer waters (Walker et al.,
2016). Near surface water was collected in a 2-L HDPE bottle and subsampling of
variables (except for *in situ* DMS, see further details below) took place on the ship
typically within 1-2h of collection. As with most sampling procedures, potential
bottle/handling effects associated with the sipper-collection method cannot be completely
ruled out. When oceanographic conditions did not permit the deployment of the sipper
(higher swell and wind speeds $> 10\ \mathrm{m\ s^{-1}}$), surface seawater samples were collected
directly from the ship with Niskin bottles mounted to a CTD rosette (water depth
corresponding to ca. 2 to 10 m on days of high wind speeds). Comparative studies
completed on surface seawater collected from both the sipper and the Niskin bottles
showed no significant differences in biological variables such as concentrations of DMS
(Walker et al., 2016). Water samples were passed gently through a 210 µm Nitex mesh
by gravity to remove large zooplankton.

Following water collection, several types of incubation experiments were conducted
onboard the ship to investigate microbial DMSP uptake and metabolism. Using the $^{35}$S-
$DMSP_d$ radiotracer approach we monitored and quantified several microbial pathways of
the degradation of $DMSP_d$ including the $DMSP_d$ loss rate constant ($k_{DMSPd}$, a measure of
the scavenging rate by bacteria of the substrate $DMSP_d$) following protocols described by
Kiene and Linn (2000b) and modifications by Slezak et al. (2007). In brief, water samples
were transferred into duplicate 71-mL dark HDPE Nalgene bottles and tracer amounts
($< 5\ \mathrm{pmol\ L^{-1}}$) of $^{35}$S-$DMSP_d$ were added to obtain a signal of ca. 1000 dpm mL$^{-1}$. Total
initial activity was first determined after gentle mixing of the bottles and subsampling of
1mL into a 10-mL scintillation vials containing 5 mL Ecolume$^{TM}$ liquid scintillation





cocktail. The bottles were then incubated for 3 h at *in situ* temperature during which time
subsamples were taken after 0, 30, 60, and 180 min to measure the loss of $^{35}$S-DMSP$_d$
over time. The $k_{DMSPd}$ was calculated as the slope of the natural log of the fraction of
remaining $^{35}$S-DMSP$_d$ versus time. Blank abiotic controls were performed at the very
beginning of the incubation experiments as well as a second time at mid-cruise using 0.2
µm-filtered seawater treated with $^{35}$S-DMSP$_d$. Loss rates in the filtered controls were
below 0.4 % of those in live samples indicating that extracellular enzyme activity was not
important in DMSP$_d$ loss.

Determination of the DMSP$_d$-to-DMS conversion efficiency (DMS yield as measured by
the recovery of $^{35}$S-DMSP$_d$ as $^{35}$S-volatiles) was conducted via parallel 24-h incubations.
Tracer amounts (< 5 pmol L$^{-1}$) of $^{35}$S-DMSP$_d$ were added to duplicate 71-mL dark HDPE
Nalgene bottles containing seawater samples in which unlabeled DMS was added at a
final concentration of 100 nmol L$^{-1}$ to allow the determination of the gross $^{35}$S-DMS
production. Initial total activity was monitored as described previously. The bottles were
incubated at *in situ* temperature for ca. 24-h, until > 90 % of the $^{35}$S-DMSP$_d$ was
consumed (Slezak et al., 2007). Upon termination of the incubation, 5 mL of sample was
transferred into a 100-mL serum vial amended with; 0.1 mL sodium dodecyl sulfate
(SDS), and 200 nmol L$^{-1}$ unlabeled DMSP$_d$ to prevent further uptake and degradation of
$^{35}$S-DMSP$_d$, and 0.05 mL Ellman's reagent (to complex thiols such as methanethiol).
Following the transfer of the samples into the serum vials, the bottles were quickly sealed
with a rubber stopper fitted with a well-cup holding a type A/E glass fiber filter soaked
with 0.2 mL stabilized H$_2$O$_2$ (3 %). The vials were set to trap the volatile $^{35}$S on an orbital
shaker and stirred at 100 rpm for ca. 6 hours (Kiene and Linn, 2000b). After trapping
was complete, the filter wicks were removed and placed in Ecolume$^{TM}$ scintillation fluid
for counting. $^{35}$S activity on the filters was considered to be $^{35}$S-DMS because the
Ellman's Reagent makes other sulfur gases (e.g. methanethiol) non-volatile. After the
volatiles were trapped, a new stopper with H$_2$O$_2$-soaked filter was placed in the vial.
Each vial was then injected with 0.2 mL NaOH (5N) through the stopper using a BD
precision guide needle to quantitatively cleave remaining $^{35}$S-DMSP$_d$ into $^{35}$S-DMS. The
$^{35}$S-DMS was trapped as described above. The DMS yield was calculated from the

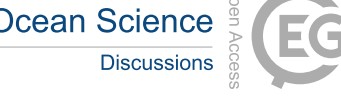



fraction of added $^{35}$S recovered as $^{35}$S-DMS in the live incubation divided by the fraction
of $^{35}$S-DMSP consumed during the incubation.

To estimate the incorporation of $^{35}$S-DMSP$_d$ into macromolecules (sulfur assimilation
efficiency), duplicate 5-mL subsamples were also taken from the previous 24-h
incubation bottles and gently filtered by manual pumping through a 0.2 µm Nylon filter
and then rinsed with trichloroacetic acid (TCA) as described in (Kiene and Linn, 2000b).
The filters were placed in 10-mL scintillation vials containing 5 mL Ecolume$^{TM}$ and the
radioactivity remaining on TCA-rinsed filters was later quantified by liquid scintillation
counting. Finally, each $^{35}$S pool measurement was expressed as a fraction of the initial
amount of added $^{35}$S-DMSP$_d$ as previously described. The measurement of the above
variables allowed us to estimate DMSP$_d$ loss rate constants (k$_{DMSPd}$), rates of gross DMS
production from DMSP$_d$ by multiplying values of k$_{DMSPd}$ with *in situ* DMSP$_d$
concentration and DMS yield. The microbial transformation rates of DMSP$_d$ measured
during these incubations are considered to stem mostly from bacterial processes however
phytoplankton-related processes cannot be totally excluded as low DMSP-producing
phytoplankton and picophytoplankton have been shown to assimilate DMSP$_d$-sulfur
(Malmstrom et al., 2005; Ruiz-González et al., 2011; Vila-Costa et al., 2006b).

Bacterial biomass production rates were measured by the incorporation of $^3$H-leucine into
TCA-insoluble. Samples were incubated in the dark for 4 h in sterile test tubes, at
ambient water temperatures and processed using standard protocols (Simon and Azam,
1989) The average CV of [$^3$H]-leucine incorporation rates for triplicate samples was ca.
10%. Rates of bacterial biomass production (µg of C L$^{-1}$ d$^{-1}$) were estimated by using a
ratio of cellular carbon to protein in bacterial cells of 0.86 (Simon and Azam, 1989).
Analysis of all radioactive samples ($^{35}$S and $^3$H) was conducted in NIWA-Hamilton (NZ)
on a Packard Tricarb liquid scintillation counter immediately following the end of the
cruise.

It has been suggested that light history and differential doses of solar radiation may
impact the growth and activity of bacteria (Herndl et al., 1993) and potentially the fate of
dissolved DMSP in seawater (Ruiz-González et al., 2012a; Slezak et al., 2001, 2007;



Toole et al., 2006). To evaluate this, we exposed near surface communities to different
light histories for 6 hours prior to $^{35}$S-DMSP$_d$ enriched bioassays: ambient variable light
(using quartz bottles in deck board incubators) or acclimation to darkness (using dark
HDPE Nalgene bottles). Rates were thus obtained during post-exposure dark incubations
(as explained above) conducted after 6 h pre-incubations at ambient light or in the dark.
Because the communities were sourced in near-surface waters during daylight hours, the
incubations conducted in quartz bottles are thought to be representative of the natural and
variable light experienced by these biological communities at the surface of the ocean.
On the whole, the light conditions (dark and ambient) at which the cells were pre-
acclimated for 6 h had no significant effect on the $^{35}$S-DMSP$_d$ metabolic rates measured.
We therefore present rate measurements made in dark-incubated samples that had been
pre-exposed to ambient light conditions for 6 h.
*3.3 Concentrations of S-compounds*
Duplicate samples of *in situ* dissolved DMSP (DMSP$_d$) and total DMSP
(DMSP$_t$ = DMSP$_p$+DMSP$_d$) were collected on board the ship using the non-perturbing
Small-Volume gravity Drip Filtration (SVDF) procedure (Kiene and Slezak, 2006). For
DMSP$_d$ samples, ca. 25 mL of seawater were gravity filtered onto GF/F and the first
3.5 mL of samples were kept in 5-mL falcon tubes amended with 50 µL 50% H$_2$SO$_4$ and
maintained in the dark at 4°C. For DMSP$_t$, 3.5 mL of unfiltered water sample were
transferred directly into 5-mL falcon tubes and treated the same way as DMSP$_d$ samples.
Subsequent analysis took place at Laval University (Canada) through alkali treatment to
cleave DMSP into DMS, purging, cryotrapping and sulfur-specific gas chromatography
(GC, see Lizotte et al. (2012)). Duplicate *in situ* DMS samples were collected directly
from the sipper or the niskin bottles by overflowing two volumes of seawater in 150 mL
crimp-top glass bottles and were analysed onboard the ship within less than 5 h of
collection following methods described in detail by Walker et al. (2016). Briefly,
calibrated volumes (5 mL) of seawater samples were purged with zero-grade nitrogen
(99.9 % pure) and gas-phase DMS was cryogenically concentrated on 60/80 Tenax TA in
a stainless steel trap at -20°C, then thermally desorbed at 100 °C for analysis by GC
coupled with sulfur chemiluminescent detection. DMS samples were also collected in 23-
mL serum vials at T0 and T6 during 6-h incubation experiments conducted in quartz



bottles on the deck of the ship (at *in-situ* light and temperature conditions) and processed
as described above.
*3.4 Statistical analysis*
Statistical analyses were carried out using the Systat statistical software for Windows
version 12.0, and Microsoft Office Excel for Mac 2011. Normality in data distribution
was determined using Kolmogorov-Smirnov tests, following which Model II linear
regressions and Spearman Rank Correlation coefficients were used to evaluate the
relationships between variables (Legendre and Legendre, 1998; Sokal and Rohlf, 1995)
Paired Student t-tests provided hypothesis assessments of the difference between
treatments.

Considering the various environmental conditions encountered during the SOAP voyage,
our dataset relied on the use of two different seawater collection approaches: the sipper
method (Walker et al., 2016) and the more standard use of Niskin bottles mounted on a
CTD rosette when periods of higher wind speeds and greater sea state prevented the
deployment of the sipper sampling equipment. Using a Wilcoxon signed-ranks test for
paired samples with non-parametric distributions, Walker et al. (2016) showed that no
significant differences ($p = 1$, $\alpha = 0.5$) were detected between samples of DMS collected
via the sipper method and those collected using Niskin bottles. This result, along with the
presence of well-mixed surface waters (MLD ranging from 14 to 40 m, Table 1) justified
the pooling of measurements made in the surface waters resulting from the two
approaches presented in the current study.

**4 Results**
*4.1 Environmental setting and biogeochemical background*
Broad scale use of ocean colour images coupled to a suite of underway sensors allowed
the successful location of three distinct blooms with varying signatures of phytoplankton
speciation and biogeochemical backgrounds (see Fig. 1, as well as (Bell et al., 2015) and
Law et al. (this issue) for further details on location of blooms and map of the cruise
track). A few general characteristics of the surface waters within sampled blooms are
presented in Table 1 to provide and overview of the oceanographic context for the 9





stations specifically sampled in this study (see Law et al. (this issue) for more detailed
description of the study area).

A first cluster of three stations was sampled between February 15[th] and 19[th] inside (sta. 1-
2) and north of (sta. 3) B1 (Fig. 1). Located in a region exhibiting Subantarctic-type
waters, B1 was characterized by the dominance of dinoflagellates (ca. 53% of total C
biomass) with *Gymnodinium* spp being responsible for an overall average of 30% of the
total dinoflagellate C biomass (Table 1). Stations 1, 2 and 3 sampled in B1 displayed an
average temperature of 14.2$^{o}$C, surface concentrations of nitrate ($NO_3^-$) ranging between
3.25 and 6.36 µmol L$^{-1}$ (mean 5.16 µmol L$^{-1}$), and concentrations of chl *a* varying from
0.91 to 1.41 µg L$^{-1}$ (mean 1.1 µg L$^{-1}$). Bacterial abundance ranged from 0.43 to 1.06 x10$^9$
cells L$^{-1}$.

The cruise track then extended further east near the Chatham Islands to capture a
coccolithophore-dominated bloom (ca. 41% of total C biomass) located in Subtropical
waters. In this area, a second cluster of three stations was sampled between February 22$^{nd}$
and 26$^{th}$ with stations 4 and 5 inside B2 and station 6 located south of B2. Temperatures
in surface waters were slightly warmer (mean 15.8$^{o}$C) than stations in B1. Stations 4 to 6
exhibited low stocks of $NO_3^-$ ranging from 0.04 to 1.32 µmol L$^{-1}$ (mean 0.5 µmol L$^{-1}$)
while near-surface concentrations of chl *a* varied between 0.53 and 1.53 µg L$^{-1}$ (mean
0.91 µg L$^{-1}$). Bacterial abundance varied between 0.59 and 1.19 x10$^9$ cells L$^{-1}$ throughout
the B2 sampling stations.

After sampling B2, the cruise path returned to the west near the first cluster of stations
sampled within Subantarctic-dominated waters. This third cluster, referred to as B3
(stations 7-9), was sampled during February 28$^{th}$ and March 5$^{th}$. Stations in B3 were
characterized   by   an   initial   mixed   phytoplankton   population   consisting   of
coccolithophores, small flagellates and dinoflagellates (B3A, Table 1) that progressively
favoured coccolithophore biomass towards the end of the sampling period (B3B). Surface
temperatures were the lowest measured during the study with a cluster average of 13.6$^{o}$C.
Surface water concentrations of $NO_3^-$ at stations 7 to 9 ranged from 2.21 to 5.28 µmol L$^{-1}$
(mean of 3.63 µmol L$^{-1}$) and concentrations of chl *a* varied between 0.39 to 0.97 µg L$^{-1}$



(mean 0.59 µg L$^{-1}$). Bacterial abundances were 0.34 and 0.51 x10$^9$ cells L$^{-1}$ at stations 8
and 9, respectively (no data is available for sta. 7, Table 1).

A transition towards deeper mixed layer depths from cluster B1 to B2 to B3 was apparent
during the sampling period; with cluster average MLD's of 15 ± 1 m, 28 ± 9 m, 37 ± 5 m,
respectively (Table 1). Trends in daily-averaged irradiance generally exhibited a decrease
between clusters with averages ranging from 263 ± 14 (W m$^{-2}$) in B1, to 251 ± 30 (W m$^{-2}$)
in B2, and finally to 192 ± 15 (W m$^{-2}$) in B3 (Table 1). Patterns of Solar Radiation
Dose (SRD) were very similar to those of daily-averaged irradiance showing a decreasing
trend from the first cluster towards the last cluster sampled.

*4.2 Reservoirs of sulfur compounds across sampling clusters*
*In situ* sea surface reservoirs of DMSP$_t$ displayed a 5-fold span across the study region
(Fig. 2a). Highest DMSP$_t$ concentrations were observed in B1, with values ranging from
118 to 160 nmol L$^{-1}$ (Fig. 2a). It is also within B1 that highest DMSP$_p$: chl *a* ratios
occurred, with a range of 89 to 141 nmol µg$^{-1}$ (Table 1). Stations sampled within B2
exhibited intermediate DMSP$_t$ pools varying from 45 to 97 nmol L$^{-1}$ and ratios of
DMSP$_p$: chl *a* that ranged from 51 to 90 nmol µg$^{-1}$ (Table 1). Surface water DMSP$_t$
concentrations within B3 were generally lower; being below 37 nmol L$^{-1}$ (sta. 7-8) but
DMSP$_t$ concentration reached 92 nmol L$^{-1}$ in the last station (sta. 9). Despite marked
differences in concentrations of DMSP$_t$ between stations 7-8 and station 9, ratios of
DMSP$_p$: chl *a* were similar within this third cluster (range of 61 to 91 nmol µg$^{-1}$, Table 1)
owing to the high chl *a* concentration measured at station 9.

Patterns of DMSP$_d$ were broadly similar to those observed for DMSP$_t$ albeit higher
variability was evident from the 18-fold difference measured between highest and lowest
concentrations (Fig. 2b). Surface seawater within sampling cluster B1 had very high
concentrations of DMSP$_d$ varying between 14 and 32 nmol L$^{-1}$. Stations sampled in B2
presented DMSP$_d$ concentrations ranging between 3 and 18 nmol L$^{-1}$. DMSP$_d$
concentrations were below 3 nmol L$^{-1}$ at stations 7-8 while DMSP$_d$ was 10 nmol L$^{-1}$ at
station 9.



Concentrations of near-surface DMS also showed high variability with a 14-fold spread
within the stations sampled (Fig. 2c). Some of the highest values of DMS were measured
in sampling cluster B1 with concentrations varying between 4.9 and 14.5 nmol DMS $L^{-1}$.
Stations 4 to 6, within the most easterly of the sampling clusters (B2) had DMS
concentrations ranging from 1 to 6.9 nmol $L^{-1}$, while stations 7-9 in B3 had a range of
DMS concentrations from 4.8 to 10.5 nmol $L^{-1}$.

*4.3 Microbial uptake and transformation of sulfur compounds*
Microbial affinity for $DMSP_d$, as indicated by the $^{35}S$-$DMSP_d$ loss rate constant ($k_{DMSPd}$;
Fig. 3a) varied between 0.4 and 3.4 $d^{-1}$, with the exception of a higher value of 19.9 $d^{-1}$
measured in the B2 cluster at station 5. The sulfur assimilatory metabolism of $^{35}S$-
$DMSP_d$, expressed as the percentage of $^{35}S$-$DMSP_d$ incorporated into macromolecules
(Fig. 3b), ranged from 1 to 4.2% across all stations. Rates of bacterial carbon production,
measured as the incorporation of $^3H$-Leucine into macromolecules, showed 5-fold
variability throughout the three sampling clusters, ranging from 0.27 to 1.46 nmol C $L^{-1}$
$d^{-1}$.

Yields of DMS from dissolved DMSP, determined as the fraction of consumed $DMSP_d$
converted into DMS, ranged from 4 to 17% (Fig. 4a), with lowest and highest yields
found within the same cluster (B3) at stations 8 and 9, respectively. The average DMS
yield in clusters B1 and B2 were very similar at 12.1% and 12.7%, respectively. The
production of DMS from $DMSP_d$, determined as the product of DMS yields and $DMSP_d$
consumption rates, varied by more than two orders of magnitude across the sampling area
(Fig. 4b). Lowest DMS production rates from $DMSP_d$ were measured in the third
sampling cluster (B3) where values remained below 0.7 nmol $L^{-1}$ $d^{-1}$. A wide-ranging set
of gross DMS production from $DMSP_d$ was estimated within B2 with 0.25 to 27 nmol $L^{-1}$
$d^{-1}$. Variability of DMS production from $DMSP_d$ within cluster B1 was lower, with rates
varying between 3.2 and 6.2 nmol $L^{-1}$ $d^{-1}$.

**5 Discussion**
*5.1 Bloom dynamics in the Subtropical Front*
The Subtropical convergence region under study was characterized by overall high



standing stocks of both autotrophic biomass (proxied by phytoplankton C and chl *a*) and
biogenic sulfur compounds (Table 1; Fig. 2a-c). The frontal zone over Chatham Rise is
known for its high productivity (Bradford‐Grieve et al., 1997; Sutton, 2001), fostering
extensive phytoplankton blooms visible from space (Sadeghi et al., 2012). Plankton
bloom dynamics are known to play a crucial role in influencing reservoirs and driving
fluxes of biogenic DMSP and DMS (Simó, 2001; Stefels et al., 2007). As evidenced by
the patterns in nutrients and chl *a*, the cruise track crossed paths with blooms in various
developmental stages in contrasting water masses. Overall quasi-depletion of silicate
standing stocks was evident from the $< 0.6$ µmol L$^{-1}$ values detected in all stations
investigated in the study region (except for sta. 6 with 1.2 µmol silicate L$^{-1}$). Nitrate
concentrations found in B1 and B3 averaged $5.2 \pm 1.7$ µmol L$^{-1}$ and $3.6 \pm 1.5$ µmol L$^{-1}$,
respectively. These nutrient signatures are a common feature of Subantarctic waters to the
South of the STF displaying depletion of silicates relative to nitrate (Sarmiento et al.,
2004). Concentrations of chl *a* in B1 (mean $1.1 \pm 0.3$ µg L$^{-1}$) were found to be higher than
a threshold concentration of 0.7 µg L$^{-1}$ used as a criterion to distinguish regions of local
biomass enrichment at the Subtropical Convergence (Llido et al. 2005). These results
coupled to the high regional phytoplankton-associated C biomass (61 µg L$^{-1}$) and the low
regional $p$CO2 minimum (260 µatm) measured in this cluster (Table1) suggests that B1
was productive and fuelled by ample nitrate reservoirs at the time of sampling. After
being away for 7 days, the cruise track returned to the Subantarctic-type waters near B1
on February 28[th] to sample the B3 cluster stations. At that time, the physicochemical and
biological signatures in B3 (sta. 7-9) differed slightly from those of B1 and displayed
higher regional $p$CO2 minimum (305 µatm), two-fold lower mean phytoplankton C
biomass (28 µg L$^{-1}$), and lower chl *a* concentrations at stations 7 and 8 (ca. 0.4 µg L$^{-1}$),
but comparable at station 9 (1 µg L$^{-1}$). Overall these results suggest that phytoplankton
biomass was lower in response to lower nutrient reservoirs and possibly greater grazing
pressure in B3, although specific information on zooplankton activity is not available.
The second cluster of stations (B2) was geographically distant from the two others (B1
and B3, Fig.1b) and had characteristics of slightly warmer Subtropical waters (Table 1).
Regionally, this study area displayed the highest $p$CO$_2$ but had similar mean
phytoplankton-associated C biomass (32 µg L$^{-1}$) to B3. Regional maximum chl *a* (max of



1.5 µg L⁻1) and nitrate levels (cluster average of $0.5 \pm 0.7$ µmol L⁻1) were the lowest
among the blooms investigated. These low nutrient features are thought to be typical of
Subtropical waters North of the Subtropical front which are also known to display
stronger vertical stratification (Llido et al., 2005). Small-celled phytoplankton (< 5µm)
are known to typically develop blooms that exhibit low chl $a$ concentrations ($< 2$ µg L⁻¹,
(Holligan et al., 1993)). Such is the case for the common and globally dominant bloom-
forming coccolithophore *Emiliania huxleyi* (Paasche, 2001) that typically has low
intracellular levels of chl $a$ ($< 0.4$ pg chl $a$ per cell, (Daniels et al., 2014)), and which
dominated the community (Law et al. this issue) during this study.

*5.2 Relating bloom dynamics with concentrations of reduced S-compounds*
Despite differences in phytoplankton dominance within blooms (Table 1), pools of
DMSP$_t$ measured in this study showed a strong association with overall phytoplankton
biomass as suggested by the positive correlation observed between DMSP$_t$ and chl $a$
($r_s = 0.83$, $p < 0.01$, $n = 9$, Table 2). A type II linear regression model suggests that 59%
of the variance in pools of DMSP$_t$ can be explained by the variability in stocks of chl $a$
(Fig. 5a). Establishing a strong relationship between DMSP and phytoplankton biomass
has historically met with limited success (Bürgermeister et al., 1990; Townsend and
Keller, 1996; Turner et al., 1988). The main reason for this being that concentrations of
DMSP are generally related to the presence of specific DMSP-rich phytoplankton species
rather than to overall phytoplankton biomass, which is often dominated by large DMSP-
poor diatoms (Lizotte et al., 2012; Stefels et al., 2007). In this study, concentrations of
DMSP co-varied significantly with phytoplankton biomass because of the persistent
dominance of dinoflagellates and coccolithophores, both DMSP-rich taxa, within the
three blooms investigated.

Unlike the strong correlation found with DMSP$_t$, no significant relationships were
detected between DMS and phytoplankton biomass (chl $a$) in our study, as reported in
Bell et al. (2015). The lack of strong relationship between DMS and chl $a$ is likely due to
many biological and physical processes involved in its production and overturning
(Dacey et al., 1998; Van Duyl et al., 1998; Kettle et al., 1999; Kwint and Kramer, 1996;





Leck et al., 1990; Scarratt et al., 2002; Simó and Pedrós-Alió, 1999; Stefels et al., 1995;
Steinke et al., 2000; Turner et al., 1988). Several studies have established links between
environmental forcings, such as the surface mixed layer depth and the irradiance regime,
and their role in driving surface DMS concentrations (Lana et al., 2012; Lizotte et al.,
2012; Miles et al., 2009, 2012; Vallina and Simó, 2007). The associations between DMS
and mixed layer depth (MLD) as well as between DMS and daily-averaged irradiance
were not found to be statistically significant within the limited dataset available in this
study ($p = 0.86$ and $p = 0.54$, respectively). Solar radiation dose (SRD) standardized over
mixed MLD was not found to improve the significance of the association between DMS
and irradiance regime. Because the spectral attenuation of solar radiation in oceanic
waters varies rapidly with depth and in association with the constituents within seawater
(Doron et al., 2007), it cannot be excluded that differences in sampling depth (sipper
versus niskin) may have obscured links between DMS and light. Heterogeneity in
sampling times (Table 1) could also have resulted in differences in light history
experienced by the DMS-producing communities. Nonetheless, DMS reservoirs and
those of its precursor DMSP were found to be abundant in the three blooming clusters as
discussed in the next section.

*5.3 High concentrations of S-compounds in Subtropical Frontal surface waters*
In this study, concentrations of $DMSP_t$ reached 110 to 160 nmol $L^{-1}$ in the first cluster, in
association with a bloom characterized by elevated concentrations of DMS (regionally up
to 20 nmol $L^{-1}$) and dominated by dinoflagellates, a diverse phytoplankton group known
for its prolific DMSP-producers (Belviso et al., 1990; Keller, 1989; Turner et al., 1988).
Few comparative DMSP datasets are available for waters near New Zealand, however the
current $DMSP_t$ concentrations are two to three times higher than the highest DMSP value
(52 nmol $L^{-1}$) reported for three open-water transects conducted between 49-76°S latitude
within the New Zealand sector of the Southern Ocean during austral spring (Kiene et al.,
2007). Species of *Gymnodinium* spp., the dominant dinoflagellate taxon in B1, have been
found to contain potentially high cytosolic DMSP (up to 244 pg DMSP/cell; (Keller,
1989)) that could have significantly contributed to the elevated reservoirs of $DMSP_t$
observed in these Subantarctic-type waters. A previous study conducted in waters of the
Subtropical Convergence Zone (40-45°S) South of Australia had demonstrated a link





between relatively high concentrations of DMSP (up to ca. 55 nmol L$^{-1}$) and
dinoflagellate biomass as well as with low microzooplankton grazing rates (Jones et al.,
1998). Gaps in the specific information concerning dinoflagellate abundance in our
sampling stations (Table 1) prevented any attempt at relating this DMSP-rich group with
overall *in situ* DMSP concentrations.

The second bloom investigated was dominated by coccolithophores and had DMSP$_t$
concentrations ranging from 45 to 96 nmol L$^{-1}$ at stations 4 to 6. *Emiliania huxleyi*, a
species exhibiting high intracellular DMSP (Franklin et al., 2010; Liu et al., 2014) and
the dominant coccolithophore in this study (Law et al, this volume), has been shown to
represent a major component of extensive coccolithophore blooms in New Zealand's
coastal waters (Chang and Northcote, 2016; Rhodes et al., 1994). Maximal
coccolithophore cell densities (up to 21.1 x10$^6$ cells L$^{-1}$) reached in the second bloom are
4 to 5-fold higher than maximal cell densities reached in coccolithophore blooms in the
North Atlantic during summer: maximum of ca. 5.5 x10$^6$ cells L$^{-1}$ (Matrai and Keller,
1993) and maximum of 4.0 x10$^6$ cells L$^{-1}$ (Malin et al., 1993) and associated with very
high levels of DMSP$_t$ (> 400 nmol L$^{-1}$). While the DMSP$_t$ concentrations were high in
B2, even higher concentrations might have been expected given the high coccolithophore
cell abundances. Variations in cell-specific DMSP quotas, nutrient and physiological
statuses of the phytoplankton communities, as well as grazing pressure (Stefels et al.,
2007) could explain these differences. *Emiliania huxleyi* is found to dominate
phytoplankton community composition in both bloom and non-bloom conditions in this
STF region (Chang and Northcote, 2016), suggesting that these relatively high summer
DMSP features could extend over a larger region which encircles the entire Southern
Ocean during austral summer in a band dubbed the "Great Calcite Belt" (Balch et al.,

591 2011).


The third and last bloom sampled (B3) was characterized by a mixed phytoplankton
population with high abundances of both dinoflagellates and coccolithophores. Although
no data for coccolithophore abundance was available at station 9, samples collected in
surface waters the day before (March 4$^{th}$) displayed coccolithophore abundance of
20.3 x 10$^6$ cells L$^{-1}$ suggesting a transition towards a coccolithophore-dominated



assemblage at the end of the sampling period. Concentrations of $DMSP_t$ (29-37 nmol $L^{-1}$)
were lower at stations 7-8 and increased to 93 nmol $L^{-1}$ at station 9, likely reflecting this
phytoplankton community shift. Pools of particulate DMSP ($DMSP_p = DMSP_t - DMSP_d$)
ranged from 26 to 83 nmol $L^{-1}$ in cluster B3 and were similar to measurements of $DMSP_p$
(ca. 28 to 40 nmol $L^{-1}$) made in waters surrounding an iron enrichment patch during the
SAGE experiment conducted in Subantarctic waters south-east of New Zealand during
the months of March and April (Archer et al., 2011). These results suggest that relatively
high concentrations of DMSP may persist in the STF zone well into the autumnal season,
which begins in mid-March in the Southern Hemisphere.

Cluster averages of DMS concentrations in this study were higher than historical data
represented in the latest DMS climatologies for the New Zealand (NEWZ) province (< 3
nmol $L^{-1}$, n = 6, Lana et al. (2011)). Clusters B1, B2 and B3 displayed average (n = 3 for
each cluster) near-surface concentrations of $9.5 \pm 4.8$, $3.6 \pm 3.0$, and $7.0 \pm 3.1$ nmol DMS
$L^{-1}$, respectively (Fig. 2c). These results underscore the fact that coverage in the previous
climatological data likely did not capture all the productive hydrographic and seasonal
features of this region. While many studies have reported on chl *a* enhancement across
frontal regions of the oceans, only a few studies have described regional increases in
DMS associated with frontal waters (Holligan et al., 1987; Matrai et al., 1996), and these
studies have provided only limited information on DMSP. Results from the current study
thus provide much needed information on the distribution of DMS but also DMSP in a
critically under-sampled area of the global ocean as well as highlight the importance of
oceanic fronts as hotspots for biogenic sulfur compounds.

Finally, an important portion of the total sea surface pools of DMSP was found as
dissolved material in this study, with 5 to 21% of $DMSP_t$ prevailing as $DMSP_d$ across the
three distinct clusters of the study region (Fig. 2b). Overall *in situ* $DMSP_d$ concentrations
ranged from 2 to 32 nmol $L^{-1}$, with highest concentrations being one order of magnitude
higher than the maximum $DMSP_d$ concentration of 2.8 nmol $L^{-1}$ found using the same
SVDF procedure by Kiene and Slezak (2006) over wide ranging ocean water types. By
examining the linear relationship between concentrations of $DMSP_p$ ($DMSP_p$ determined
as $DMSP_t - DMSP_d$) and those of $DMSP_d$ (Fig. 5b) we are able to show that the slope





(0.21) of the Model II regression analysis is very similar to the slope (0.20) obtained by
Kiene and Slezak (2006) for SVDF $DMSP_d$ samples from the Sargasso Sea. Although it
is impossible to entirely circumvent bottle, filtration and/or processing effects that could
lead to overestimation of $DMSP_d$ concentrations, despite careful handling, it is
nonetheless noteworthy that, despite large contrasts in trophic status, our results show a
tendency for $DMSP_d$ to build up in surface waters in proportion to its particulate
counterpart, constituting up to 21% of the total DMSP pool in our study. The fuelling of
dissolved DMSP reservoirs in the water column has biogeochemical importance
considering this compound supplies heterotrophic micro-organisms with C and S as is
discussed in the next section.

*5.4 Cycling of S-compounds through heterotrophic bacterioplankton*
*5.4.1 Wide-ranging microbial $DMSP_d$ rate constants*
To our knowledge, this study provides the first DMSP process rate measurements across
a frontal zone, within three quasi co-occurring but distinct phytoplankton blooms. Except
for station 5, which will be discussed below, $DMSP_d$ loss rate constants ($k_{DMSPd}$) varied
between 0.4 and 3.4 $d^{-1}$, suggesting wide-ranging turnover times of $DMSP_d$ reservoirs,
between ca. 7 hour to 2.5 days (Fig. 3a). Assuming steady state conditions, these turnover
times imply that between ca. 2 to 14% of the DMSP stock was renewed hourly by
autolysis, exudation viral attack and grazing (Stefels et al., 2007). These results are
comparable with similar ranges of $k_{DMSPd}$ measurements conducted in various oceanic
environments (Table 3). Our highest value of $k_{DMSPd}$ (19.9 $d^{-1}$) was recorded at station 5,
within B2. High $k_{DMSPd}$ values are not commonly reported in the literature except for the
22.1 $d^{-1}$ observed by Royer et al. (2010) in the NE Pacific which was similar to our
highest rate. These very rapid turnover times (ca. 1 hour at sta. 5) could reflect transient
periods of increased bacterial abundance or production. *In situ* rates of leucine
incorporation by bacteria were not particularly high at station 5 (0.62 compared to an
overall range of 0.27 to 1.46 nmol $L^{-1}$ $d^{-1}$) nor was the abundance of heterotrophic
bacterial cells (0.85 at sta. 5, range of 0.34 to 1.19 x $10^9$ cells $L^{-1}$) and the concentration
of $DMSP_d$ (9 compared to a global range of 2 to 32 nmol $L^{-1}$). Furthermore, in our study
no overall significant trends were detected between $DMSP_d$ loss rate constants ($k_{DMSPd}$)
and numbers of bacteria or rates of leucine incorporation. It has been suggested that loss

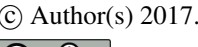



rate constants of $DMSP_d$, rather than being directly related to stocks of bacteria could be
more related to bacterial community composition, and particularly the specific abundance
of Roseobacter, a member of Alphaproteobacteria, and with Gammaproteobacteria
(Royer et al., 2010), which are both significant contributors to DMSP metabolism
(Malmstrom et al., 2004a, 2004b; Vila-Costa et al., 2007; Vila et al., 2004). On the
whole, microbial $DMSP_d$ rate constants were variable within the study region (50-fold
range), with no specific responses related to the presence of diverging phytoplankton
assemblages and biological characteristics within blooms.

*5.4.2 Fulfilled bacterial sulfur requirements in a sulfur-rich environment*
The assimilatory metabolism of sulfur from DMSP is a key control on the amount of this
compound diverted away from DMS. Assimilation efficiency of sulfur from $^{35}S$-$DMSP_d$
into bacterial macromolecules was low (< 5%) throughout the study region (Fig. 3b).
Values reported in this study are below a relatively narrow range of DMSP-S assimilation
efficiency values reported in various studies (see Table 3). Taking into account the
DMSP-S incorporation efficiency, the potential contribution of DMSP-S to bacterial
sulfur biomass production was estimated from bacterial C production and lower and
upper limits of bacterial C:S molar ratios (32 to 248 from (Cuhel and Taylor, 1981;
Fagerbakke et al., 1996). For all the reported C:S values, calculated DMSP-S
incorporation exceeded 100% of bacterial sulfur biomass production estimates (data not
shown) suggesting that DMSP availability was in excess of bacterial sulfur requirements.
These results agree with several studies (Kiene and Linn, 2000b; Simó et al., 2009; Vila-
Costa et al., 2007, 2014) suggesting that DMSP acts as a major source of S for
heterotrophic bacterioplankton. A possible caveat of these estimates is the fact that
DMSP-S assimilation includes that which might be taken up by cyanobacteria and
phytoplankton (Malmstrom et al., 2005; Vila-Costa et al., 2006a), which likely don't
contribute to leucine incorporation. This would lead to overestimation of the contribution
of DMSP to bacterial S production. Overall, and assuming that heterotrophic bacteria
dominate the uptake of DMSP, the S assimilation efficiencies (< 5%) measured in this
study point towards a rapid saturation of S requirements by the microbial assemblages in
DMSP-rich waters of the Subtropical Front.





### 5.4.3 Microbial DMS yield and gross production of DMS from $DMSP_d$

Microbial DMS yields, the conversion efficiency of $DMSP_d$ into DMS, varied from 4 to 17% with an overall average of 11% across the entire study region, irrespective of water mass provenance and bloom association (Fig. 4a). Our results add to the mounting evidence that, as a whole, the span in endogenous proportions of $DMSP_d$ consumed by bacteria and cleaved into DMS is similar across various oceanic environments (see Table 3). A significant and positive relationship was found between rates of bacterial leucine incorporation and DMS yields in this study ($r_s = 0.84$, $p < 0.01$, $n = 8$). This relationship suggests that as carbon incorporation for protein synthesis was heightened in the microbial communities, the proportional use of DMSP as a carbon source also increased, leading to higher $DMSP_d$-to-DMS conversion efficiencies (Table 2). Furthermore, prokaryotic protein synthesis, estimated by the bacterial incorporation of leucine (Kirchman et al., 1985), appeared to be significantly associated with the supply of $DMSP_d$ in this study ($r_s = 0.86$, $p < 0.01$, $n = 8$, Table 2). The fate of S in DMSP-metabolizing bacterial communities is complex and most likely affected by numerous factors, at least one of which is the S requirement relative to the availability of organic S. Findings from this study are consistent with the hypothesis that organic S in excess of bacterial requirements biases DMSP metabolism against demethylation (Kiene et al., 2000; Levasseur et al., 1996; Pinhassi et al., 2005). These observations agree with results from Lizotte et al. (2009) who observed an increase in DMS yields following the addition of non-limiting concentrations of $DMSP_d$ and increases in microbial incorporation of leucine during an Ocean Iron Fertilization experiment in the Subarctic Pacific. Furthermore, at a physiological level, factors including bacterial carbon requirements and concentrations of DMSP degradation products can also exert an impact on the fate of DMSP (Kiene et al., 2000). Since the radioisotope technique used to examine the microbial cycling of $DMSP_d$ traces only the S moiety, significant respiration of C-DMSP can occur (Vila-Costa et al., 2010). As such, the combination of rather typical $DMSP_d$ turnover times (overall average of < 1 day) and low DMSP-S assimilation efficiencies (< 5%) could be an indication of the availability of C-rich compounds, including DMSP, to the bacterial assemblages in this study.





Regardless of the positive associations between bacterial carbon production and the
supply of $DMSP_d$, as well as $DMSP_d$ conversion efficiency into DMS, yields of DMS
never exceeded 17%. Altogether, our results reinforce the concept that DMSP-to-DMS
conversion is not the main fate of microbial $DMSP_d$ turnover in natural environments (see
reviews by Simó (2001) and Stefels et al. (2007)), never exceeding 31% of consumed
$DMSP_d$ in most $^{35}$S-DMSP tracer studies (see compilation in Table 3). However, even
modest variance in $DMSP_d$-to-DMS conversion efficiencies can result in considerable
variations in the production rate of DMS in sea surface waters. In this study, gross DMS
production from $DMSP_d$ ranged from near detection limits to a high of 27 nmol of DMS
per liter per day (Fig. 4b).  The latter estimate resulted from high $DMSP_d$ loss rate
constant coupled to high $DMSP_d$-to-DMS conversion efficiency at station 5 (Fig. 3a, Fig.
4a). Omitting this very high rate measured on February 24[th], DMS production from
$DMSP_d$ contributed on average 2.3 nmol $L^{-1}$ $d^{-1}$ of DMS to near surface reservoirs
(ranging from 0.07 to 6.2 nmol DMS $L^{-1}$ $d^{-1}$) of the study region. These values are
comparable to DMS production rates from $DMSP_d$ previously reported (Table 3). It is
noteworthy that although production rates of DMS from $DMSP_d$ were low in B3,
concentrations of DMS remained high despite slightly higher wind speeds during this
period of sampling (see Bell et al. (2015)), which should have enhanced ventilation of
DMS to the atmosphere. This suggests that sinks for DMS were somehow alleviated, for
example through: (1) a decrease in photo-oxidation of DMS related to a reduction in
irradiance fields and a deepening of the mixed layer (see Table 1); (2) a reduction in
bacterial consumption of DMS, for which unfortunately no specific information is
available but that could be associated with a decrease in bacterial abundance (Table 1).

Alternatively, but not excluding these potential sinks, other sources of DMS (non-
bacterial) are likely to have contributed to the concentrations of DMS. Assuming steady-
state conditions, the comparison between our microbially-mediated DMS production
rates and the concentrations of DMS in near-surface waters suggest that bacteria alone
could not have sustained the DMS pool at most stations, and particularly in B3. Average
calculated DMS turnover times due to production from $DMSP_d$ were similar between B1
(2.3 days) and B2 (2.4 days) but increased to an average 36.5 days in B3. Considering
that DMS sinks commonly proceed on time scales of hours to a few days (Simo et al.,



2000; Stefels et al., 2007), the lengthier bacterial DMS turnover times in B3 point
towards the importance of community-associated DMS production in fuelling DMS in
surface waters. Community DMS production may have included indirect processes such
as zooplankton grazing, viral lysis, and senescence, as well as direct algal DMSP-lyase
activity associated with the presence of certain species of dinoflagellates and
coccolithophores (Niki et al., 2000; Wolfe and Steinke, 1996), ubiquitous in Subantarctic
waters in early March. Another indication of the relative importance of phytoplankton-
mediated DMS production in B3 stations can be found in the comparison of standing
stocks of DMS relative to $DMSP_t$ which averaged 0.07 and 0.05 mol:mol in B1 and B2,
respectively, and increased to a mean of 0.15 mol:mol in B3. This higher average
$DMS:DMSP_t$ molar ratio suggests stronger $DMSP_p$ to DMS conversion efficiency in this
particular sampling cluster. Further, albeit limited, information on net community-
associated DMS production is provided by net changes in DMS concentrations (Fig. 6)
calculated as the difference between concentrations at the beginning and at the end of the
6-h pre-acclimation incubations under *in-situ* light conditions. These net changes include
all sources and sinks of DMS (except for ventilation). Net changes in DMS
concentrations over the 6-h period showed overall accumulation of DMS in the
incubation experiments (maximum of 10.8 nmol $L^{-1}$ at sta. 9 in B3). An exception to the
accumulation trend was seen at station 8 where a net consumption of DMS
(-1.1 nmol $L^{-1}$) took place over the 6-h incubation at station 8. Coarse calculations that
assume steady-state conditions suggest that transposing these net changes over a daily
period amounts to a mean net community production of DMS from $DMSP_t$ of
15.2 nmol $L^{-1}$ $d^{-1}$ (n = 6) throughout the stations where data was available. This rough
estimate is almost 3 times as high as the gross microbial production of DMS from
$DMSP_d$ (average of 5.3 nmol $L^{-1}$ $d^{-1}$, n = 6) in the same stations (sta. 3, 5, 6, 7, 8 and 9).
The microbial DMS production rates from $DMSP_d$ in this study are also considerably
lower than several of the community net production rates required to support microlayer
DMS (range of -1445 to 5529 nmol $L^{-1}$ $h^{-1}$) reported by Walker et al. (2016). Altogether
our findings support the view that indirect and direct processes of phytoplankton-
mediated DMS production were important contributors to standing stocks of DMS in the
near-surface waters of the STF during austral summer.



## 6 Conclusions

Our study provides much needed information on both concentrations and cycling of dimethylated sulfur compounds within waters of the New Zealand biogeochemical province (NEWZ) and more specifically in an oceanic frontal region. The three distinct phytoplankton blooms sampled were shown to be hotspots for concentrations of DMS (max of 14.5 nmol $L^{-1}$) and $DMSP_t$ (max of 160 nmol $L^{-1}$). Regardless of physico-chemical and biological differences in bloom dynamics across the Subantarctic and Subtropical waters investigated, pools of $DMSP_t$ varied in concert with stocks of chl $a$, likely because of the dominance of DMSP-rich phytoplankton groups such as dinoflagellates and coccolithophores. The significant relationship between chl $a$ and $DMSP_t$ ($r_s = 0.83$, $p < 0.01$) across blooms suggests that autotrophic biomass may be a reasonable predictor of DMSP for this region during austral summer. The high availability of reduced sulfur fully satisfied sulfur requirements of the micro-organisms leading to overall low microbial sulfur assimilation efficiencies from $DMSP_d$ ($< 5\ \%$). Microbial yields of DMS varied 4-fold over the Subtropical Front (4-17 %) and were significantly correlated with bacterial protein synthesis rates, lending support to the idea that supplies of $DMSP_d$ were non-limiting. Microbially-mediated DMS production from $DMSP_d$ generally ranged between 0.1 to 6.2 nmol DMS $L^{-1}$ $d^{-1}$, but was as high as 27 nmol DMS $L^{-1}$ $d^{-1}$ at station 5. The comparison between standing stocks of DMS and microbially-mediated DMS production rates suggest that bacteria alone could not have sustained DMS concentrations in near-surface waters at most stations in this study. These results point towards phytoplankton-associated production of DMS as an important co-driver of DMS pools in the surface waters on either side of the STF. While the STF was already a known region of high biological activity, results from the current study reinforce the hypothesis that the STF also supports high DMSP-to-DMS conversions largely related to its abundant biogenic sulfur compounds. These findings could have important implications for global sulfur budgets and climate considering that the STF covers several hundred kilometers in a ring encircling a part of the globe with little anthropogenic influence, and where productive plankton blooms may persist over several months



**7 Acknowledgements**

We thank Captain Evan Solly and the entire crew of the R/V Tangaroa; Els Maas for facilitating radio-isotope work during the research cruise; F. Hoe Chang for coccolithophore abundance data, Anathea Albert for $^{35}$S-DMSP scintillation counts; Timothy Burrell and Karen Thompson for bacterial production scintillation counts, Matt Walkington for irradiance data processing and validation, as well as CTD operations, Marieke van Kooten for nitrate measurements, and Murray Smith for MLD calculations. This paper is a contribution to the research programmes of Québec-Océan and the Biology Department of Laval University as well as to the New Zealand Surface Ocean Lower Atmosphere Study (SOLAS). This study was supported by funding from NIWA's Climate and Atmosphere Research Programme 3 – Role of the oceans (2015/16 SCI), and a Postdoctoral Fellowship (CO1X0911) for CW from the New Zealand Ministry for Business, Innovation and Employment (MBIE). RP Kiene acknowledges support from the National Science Foundation, grants OCE-0928968 and OCE-1436576.

**8 Author contribution**

M. Lizotte, M. Levasseur designed the experiments and M. Lizotte, C. S. Law, C. F. Walker, K. A. Safi, and A. Marriner carried out the experiments and performed the measurements in the field. R. P. Kiene produced and provided $^{35}$S-DMSP$_d$ for the radiotracer experiments. M. Lizotte prepared the manuscript with contributions from all co-authors.

**9 Competing interests**

The authors declare that they have no conflict of interest.

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





**11 Figures**

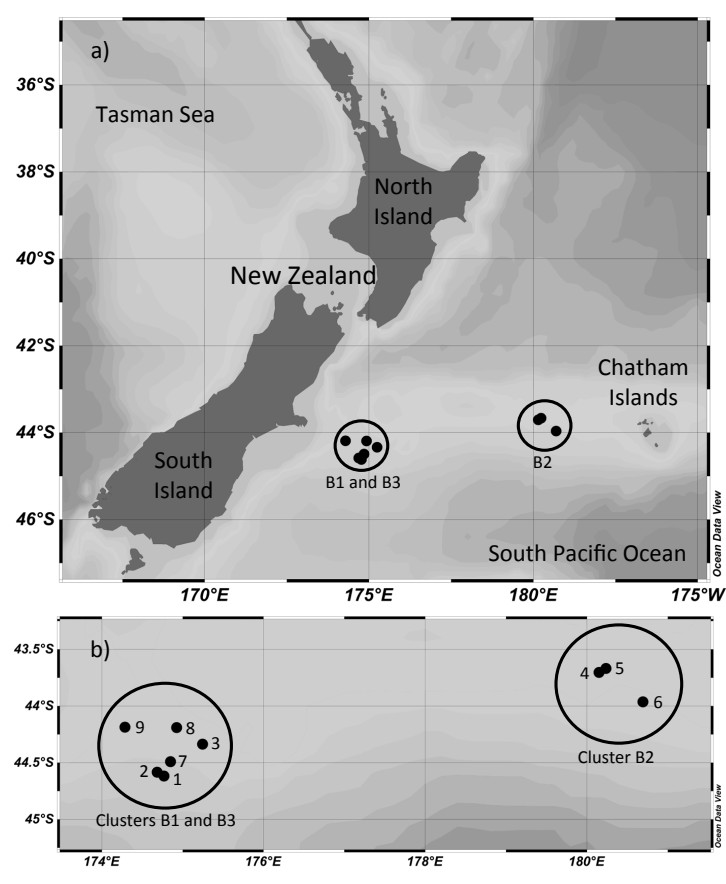


Figure 1. (a) Map of the general sampling area over the Chatham Rise East of New
Zealand's South Island; and (b) close-up of the partitioning of the 9 stations in clusters
B1, B2 and B3 sampled during the SOAP voyage in February and March 2012.











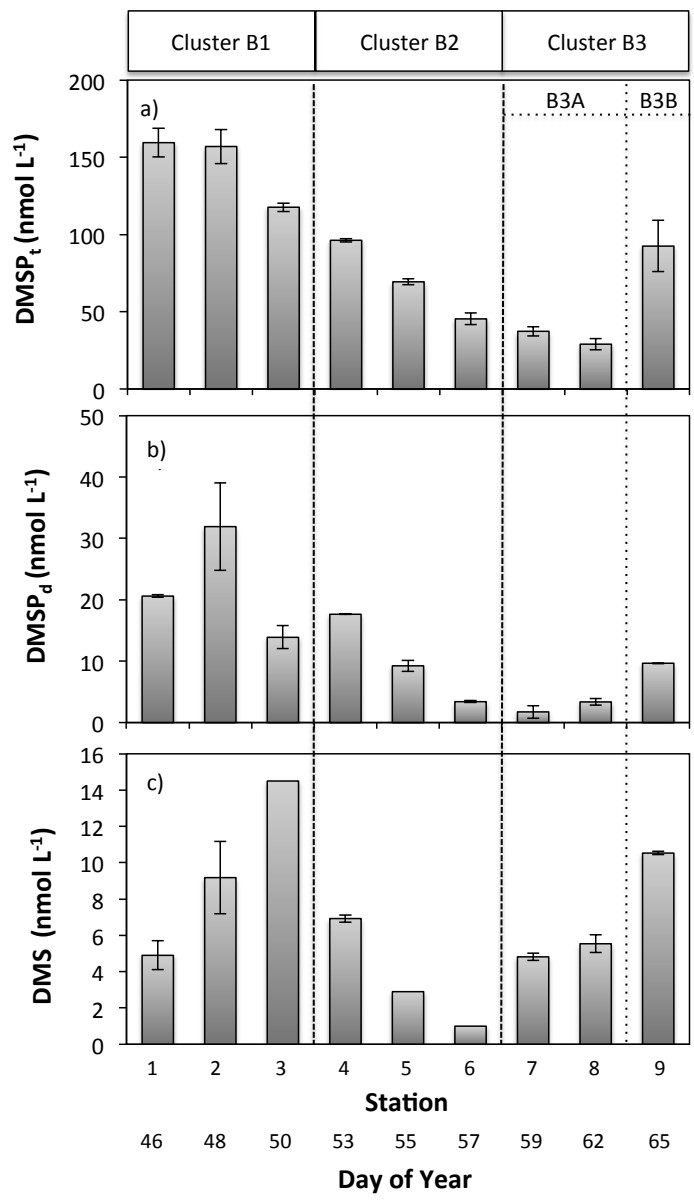


Figure 2. Concentrations of (a) total DMSP (DMSP$_t$); (b) dissolved DMSP (DMSP$_d$); and
(c) DMS measured at nine stations during the SOAP voyage in February and March
2012. Values are means of experimental duplicates and error bars represent the absolute
deviations of data points from their mean. DMS data from stations 3,5 and 6 represent
single samples, while values from stations 7 and 8 come from matching T0 DMS values
(from incubation experiments). The three sampling clusters are noted as B1, B2, and B3.




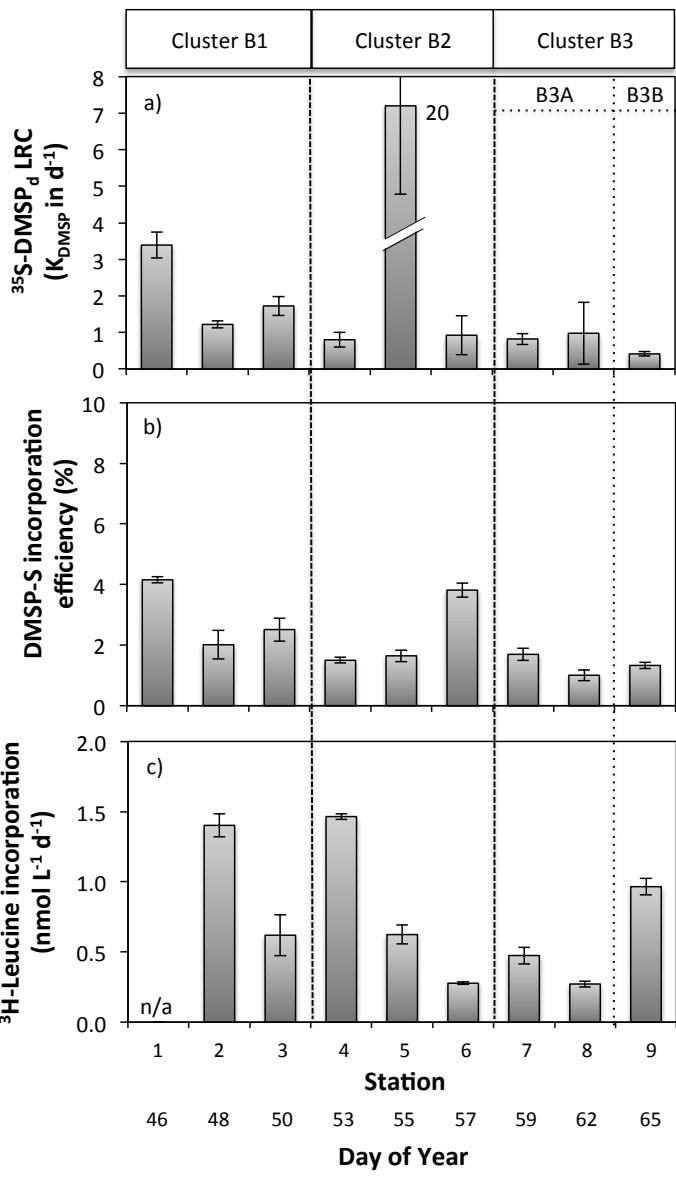


Figure 3. (a) Microbial $DMSP_d$ loss rate constant ($k_{DMSPd}$ in $d^{-1}$); (b) Microbial
assimilation efficiency of DMSP-S into macromolecules (%); (c) Microbial $^3$H-Leucine
incorporation (nmol $L^{-1}$ $d^{-1}$) at nine stations during the SOAP voyage in February and
March 2012. The three sampling clusters are noted as B1, B2, and B3. Stacks and error
bars indicate mean and standard deviation of triplicate samples. n/a = not available.






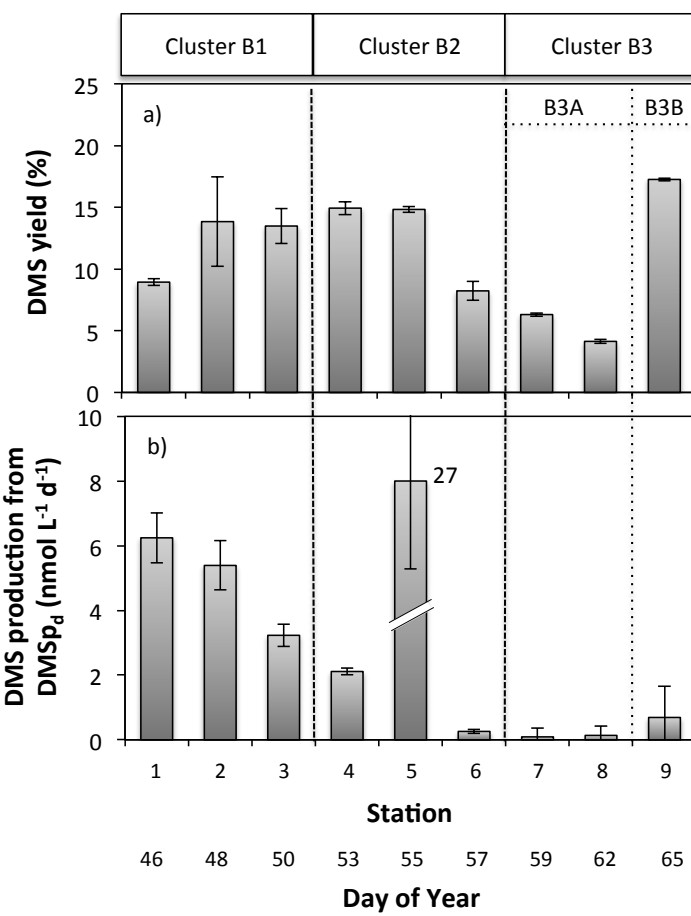

Figure 4. (a) Microbial DMS yields (%); (b) Gross DMS production from DMSP$_d$ (nmol L$^{-1}$ d$^{-1}$) at nine stations during the SOAP voyage in February and March 2012. The three distinct sampling clusters are noted as B1, B2, and B3. Stacks and error bars indicate mean and standard deviation of triplicate samples.



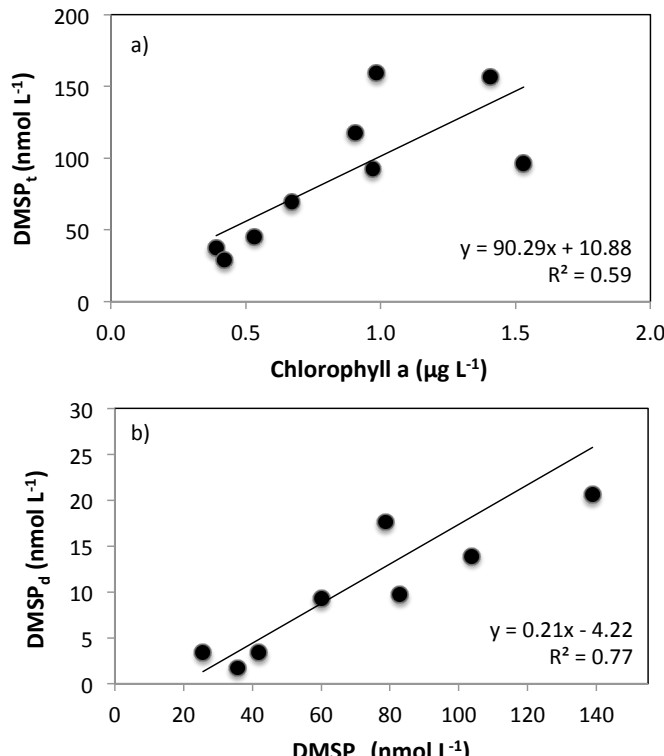


Figure 5. Model II regressions between (a) concentrations of chl *a* and DMSP$_t$; (b)
concentrations of DMSP$_d$ and DMSP$_t$.














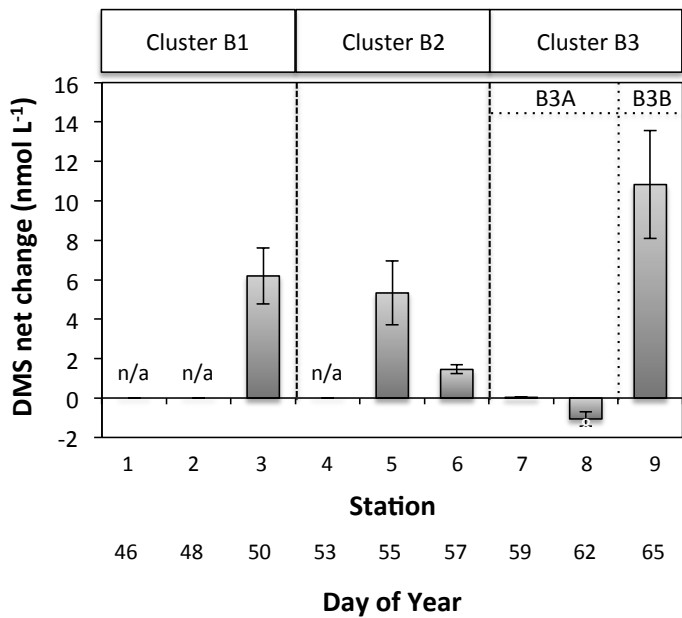

Figure 6. Net changes in DMS concentrations calculated as the difference between T0 and T6 values during 6-h incubation experiments conducted in quartz bottles (at *in situ* light and temperature conditions) on the deck of the ship during the SOAP voyage in February and March 2012. Stacks and error bars indicate mean and standard deviation of triplicate samples. n/a = not available.



**12 Tables**

Table 1. Broad biogeochemical characteristics of the stations sampled within three blooms during the SOAP voyage in February and March 2012.

| | Bloom 1 | | | Bloom 2 | | | Bloom 3 | | |
|---|---|---|---|---|---|---|---|---|---|
| Regional $pCO_2$ min (µatm) | 260 | | | 339 | | | 305 | | |
| Regional Chl$a$ max (µg L$^{-1}$) | 5 | | | 1.5 | | | 3.5 | | |
| Regional DMS max (nmol L$^{-1}$) | 20 | | | 15 | | | 10 | | |
| Regional mean phytoplankton C biomass (µg L$^{-1}$) | 61 | | | 32 | | | 28 | | |
| | Cluster B1 | | | Cluster B2 | | | Cluster B3A | | Cluster B3B |
| Regional predominant phytoplankton (of C biomass) | Dinoflagellates | | | Coccolithophores | | | Mixed population | | Coccolithophores |
| Day of Year | 46 | 48 | 50 | 53 | 55 | 57 | 59 | 62 | 65 |
| Date in 2012 | 15 February | 17 February | 19 February | 22 February | 24 February | 26 February | 28 February | 02 March | 05 March |
| Sampling time (NZST) | 8h05 | 8h02 | 7h30 | 8h27 | 7h00 | 6h52 | 7h30 | 8h00 | 9h04 |
| Sampling coordinates | 44°37.3'S | 44°35.2'S | 44°20.7'S | 43°42.9'S | 43°40.4'S | 43°57.44'S | 44°29.27'S | 44°11.23'S | 44°11.10'S |
| | 174°46.3'E | 174°41.4'E | 175°14.45'E | 179°51.6'W | 179°45.56'W | 179°18.30'W | 174°50.56'E | 174°55.28'E | 174°17.7'E |
| Location in relation to bloom | In Bloom 1 | In Bloom 1 | N of Bloom 1 | In Bloom 2 | In Bloom 2 | S of Bloom 2 | In Bloom 3 | In Bloom 3 | In Bloom 3 |
| Sequential station number | 1 | 2 | 3 | 4 | 5 | 6 | 7 | 8 | 9 |
| Predominant water mass | SAW | SAW | SAW | STW | STW | STW | SAW | SAW | SAW |
| Sampling depth (m) | 1.6 | 1.6 | 2 | 1.6 | 2 | 2 | 10* | 10* | 1.6 |
| Mixed layer depth (m) | 14 | 14 | 16 | 21 | 39 | 25 | 31 | 39 | 40 |
| Daily averaged irradiance (W m$^{-2}$) | 258 | 279 | 252 | 222 | 249 | 282 | 181 | 185 | 208 |
| Solar Radiation Dose (W m$^{-2}$) | 90 | 79 | 79 | 40 | 39 | 75 | 41 | 39 | 26 |
| Silicate (µmol L$^{-1}$) | 0.40 | 0.39 | 0.34 | 0.22 | 0.40 | 1.16 | 0.22 | 0.58 | 0.18 |
| Nitrate (NO$_3^-$ µmol L$^{-1}$) | 6.36 | 3.25 | 5.86 | 0.04 | 1.32 | 0.13 | 2.21 | 5.28 | 3.41 |
| Chl$a$ (µg L$^{-1}$) | 0.99 | 1.41 | 0.91 | 1.53 | 0.67 | 0.53 | 0.39 | 0.42 | 0.97 |
| Bacteria (*10$^9$ cells L$^{-1}$) | 1.06 | 0.69 | 0.43 | 1.19 | 0.85 | 0.59 | n/a | 0.34 | 0.51 |
| Coccolithophores (*10$^6$ cells L$^{-1}$) | 1.19¶ | 9.46¶ | 5.19 | 12.70 | 5.80 | 21.13 | 4.68 | 3.90 | n/a§ |
| DMSP$_p$:Chl$a$ ratio (nmol µg$^{-1}$) | 141 | 89 | 115 | 51 | 90 | 79 | 91 | 61 | 85 |

Regional data represents maxima/minima or averages in the surface waters within blooms and encompass more stations then the 9 presented specifically in this study (See Law et al., this issue).
SAW (Subantarctic Water) STW (Subtopical Water). Data that is not available = n/a. *Prevailing high windspeeds (>10 m s$^{-1}$) and heavy seas prevented the sampling of near surface samples at these stations.
¶Values from matching CTD data at 2 m. §No coccolithophore data is available for this date, however samples taken on March 4$^{th}$ showed coccolithophore abundance of 20.3 *10$^6$ cells L$^{-1}$.












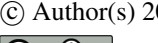




Table 2. Spearman's rank correlation coefficients ($r_s$) for various variables measured
during SOAP.

| Variables | | $r_s$ coefficient |
|---|---|---|
| Chl *a* | $DMSP_t$ | 0.83** |
| $DMSP_p$ | $DMSP_d$ | 0.92*** |
| Leucine incorporation | $DMSP_d$ | 0.86** |
| Leucine incorporation | DMS yield | 0.84** |

***$p < 0.001$ and **$p < 0.01$, n = 9 for all variables except for leucine incorporation where n = 8.



Table 3. Partial compilation of microbial DMSP$_d$ and DMS cycling rates measured via the $^{35}$S radioisotope technique in papers published since 2000.

| Study | Area of study | Time of year | Particularities | Sampling depth (m) | Temperature (°C) | Endogenous DMSP$_d$ (nmol L$^{-1}$) | DMSP$_d$ loss rate constant $k_{DMSPd}$ (d$^{-1}$) | DMSP$_d$ turnover time (d) | DMSP$_d$ turnover rate* (nmol L$^{-1}$ d$^{-1}$) | Sulfur assimilation efficiency** (%) | DMS yield (%) | DMS production from DMSP$_d$ (nmol L$^{-1}$ d$^{-1}$) |
|---|---|---|---|---|---|---|---|---|---|---|---|---|
| Kiene & Linn 2000a | Northern Gulf of Mexico | September 1997 (Late summer) | Coastal and oceanic waters | 1 - 100 | 22 - 30 | 0.2 - 10 | n/a | 0.03 - 0.6 (range of means) | 0.3 - 129 | 5 - 40 | n/a | 0.2 - 5.9 (range of means) |
| Kiene & Linn 2000b | Subtropical northern Gulf of Mexico, Northern Sargasso Sea, temperate North Atlantic. | Sept. 1997 to Jan 1999 (4 seasons) | Coastal and oceanic waters | 0 - 95 | 3 - 28 | 1 - 4 | n/a | n/a | n/a | n/a | 2 - 21 | n/a |
| Zubkov et al. 2002 | Northern North Sea | June 1999 (Summer) | Lagrangian SF$_6$ tracer study of a E. huxleyi bloom | 2 - 50 | 8.5 - 11.5 | 8.0 ± 3.6 (in patch) 10.1 ± 5.7 (out patch) | n/a | 0.1 - 0.4 (in patch) 0.2 - 0.3 (out patch) | 20 ± 8 (in patch) 21 ± 5 (out patch) | 2.5 ± 1.3 (in patch) 2.0 ± 0.8 (out patch) | 6 - 12 | 2 - 2.5 |
| Pinhassi et al. 2005 | Coastal Gulf of Mexico | June 2001 (Summer) | Microcosm experiment (only controls shown) | 0.5 | 27 | 3 - 6 | 5 - 15.1 | 0.1 - 0.2 | n/a | 29 | n/a | n/a |
| Merzouk et al. 2006 | Subarctic NE Pacific | July 2002 (Summer) | HNLC waters outside an iron-enriched patch | 1 - 14 | n/a | 2.8 - 19 | 1.3 - 6.2 | 0.2 - 0.6 | 4.8 - 72 | n/a | n/a | n/a |
| Kiene et al. 2007 | New Zealand sector of Southern Ocean | November 2003 & 2005 - December 2004 (Spring-summer) | Presence of ice along transects | 2 - 4 | -1.8 - 8.7 | < 4 | n/a | n/a | 0 - 12.5 | n/a | n/a | n/a |
| Merzouk et al. 2008 | Northwest Atlantic | April-May 2003 (Spring) | Senescent diatom bloom | 10 | 2.6 - 3.4 | 0.7 - 3.9 | 1.7 - 13 | 0.1 - 0.6 | 5 - 28 | n/a | 9 - 18 | 0.5 - 2.4 |
| Vila-Costa et al. 2008 | Coastal Mediterranean Sea (Blanes Bay) | January 2003 to June 2004 | Seasonal survey, shallow water column (24m) | 0.5 | 12.8 - 24.6 | 5 ± 2 | 0.8 - 6.3 | 0.2 - 1.3 | 2 - 24 | n/a | 3 - 37 | 0.1 - 7.7 |
| Simo et al. 2009 | Coastal Mediterranean Sea (Blanes Bay) | January 2003 to March 2004 | Seasonal survey, shallow water column (24m) | 0.5 | 11 - 25.2 | n/a | n/a | n/a | 2 - 24 | 1 - 46 | n/a | n/a |
| Lizotte et al. 2009 | Subarctic NW Pacific | July-August 2004 (Summer) | HNLC waters outside an iron-enriched patch | 5 | 8.3 - 11.9 | n/a | n/a | n/a | n/a | 18 - 25 | 7 - 13 | n/a |
| Royer et al. 2010 | Subarctic NE Pacific | May-June 2007 (Early summer) | Along a natural iron gradient from coastal to open waters | 10 | 7.1 - 11 | 1.3 - 3.6 | 2.1 - 22.1 | 0.1 - 0.3 | 8.6 (mean offshore) 42 (mean inshore) | 10 - 29 | 3 - 13 | 0.7 (mean offshore) 1.6 (mean inshore) |
| Luce et al. 2011 | Canadian Arctic Archipelago | October - November 2007 (Late fall) | 20 Stations from Northern Baffin Bay to the Beaufort Sea through the Northwest Passage | 2 - 3 | -1.8 - 0.1 | 0.1 - 5 | 0.2 - 3.4 | 0.3 - 4.1 | 0.2 - 5.8 | n/a | 4 - 15 | 0.01 - 0.5 |
| Lizotte et al. 2012 | Northwest Atlantic | May-July-October 2003 (3 seasons) | Seasonal survey of 7 biogeochemical provinces | 8 - 15 | 2 - 26 | 0.5 - 9 | 0.7 - 4.1 | 0.2 - 1.4 | 0.3 - 24.3 | n/a | 3 - 21 | 0.01 - 3.1 |
| Motard-Côté et al. 2012 | Canadian Arctic Archipelago | September 2008 (Fall) | Northern Baffin Bay/Lancaster Sound | 5 | -1.3 - 3.8 | n/d - 2.1 | 0.7 - 2.6 | 0.4 - 1.4 | n/a | 11 - 18 | 12 - 31 | n/a |
| Vila-Costa et al. 2014 | Bermuda Atlantic Time-series Study (BATS) station | September 2007 (Fall) | Short-term enrichment studies (organic substrates enrichments) | 10 | 27.5 | 5.9 ± 0.8 | n/a | n/a | 2.6 - 28.5 | 3 - 23 | 1 - 45 (control < 20) | n/a |
| This study | New Zealand Subtropical Front | February-March 2012 (Late summer) | Frontal zone (Subantarctic and Subtropical water masses) | 1.6 - 10 | 13.5 - 15.7 | 1.7 - 31.9 | 0.8 - 19.9 | 0.1 - 1.6 | 1.4 - 184 | 1 - 4 | 4 - 17 | 0.1 - 27.3 |

*Also called the microbial DMSP$_d$ consumption rate. **Measured from the incorporation of $^{35}$S into TCA-insoluble particles. Expressions n/a and n/d refer to data that is non-available and non-detectable, respectively. This compilation is non-exhaustive and does not include certain stressor experiments for simplicity (see additional studies including Slezak et al. 2007; Ruiz-Gonzalez et al. 2011; 2012a; 2012b).