# Peer review of "Dimethylsulfoniopropionate (DMSP) and dimethylsulfide (DMS) cycling"

_Ocean Science, 2017_

## Referee Comment (RC1) · M. Vila-Costa (Referee) · 21 Jun 2017

This is a very well written paper about DMS(P) concentrations and dynamics in one rather unexplored oceanic site, the frontal region above the Chatam Rise east of NZ. The authors measured [DMS] and [DMSP] concentrations and microbial DMSP turnovers over a relatively short period of time, sampling 3 different blooms of DMSP-producing phytoplankton. Due to the non-commercial nature of 35S-DMSP, measurements of microbial DMSP turnover, fates and DMS yields are rare in the literature. This paper provides useful data for DMSP modelers since it covers a poorly characterized

zone although very active in terms of use of reduced sulfur in the ocean.

The methodology used is correct and well described. As a general comment, the only weakness detected on this study is that not all pools of DMS(P) cycling were covered since no measurements of DMSO were performed (particulate and dissolved) which hampers a more extended discussion on the fate of metabolized DMS in seawater.

It is really appreciated negative results of influence of light preincubations on DMSP dynamics. I think it is not stressed enough in the discussion of the paper. One thinks it is a pity than in such DMSP-active zone more specific experiments to test still open questions of the cycle, mainly related to the different physiological and ecological roles of DMSP in the upper ocean could have been tested (for instance, the relative role of non-DMSP-producers algae as sink of DMSP, algal DMS production, new in situ production of DMSP by heterotrophic bacteria, chemotaxis, etc). Rather than a weakness, I hope the paper will encourage the DMSP community to sample in the described area.

I only have minor comments on the manuscript. line 38: there is more than only 2 fates of consumed DMSP, excretion as an oxidized form but not incorporated into cell structure is missed. line 45. "measured in this study" can be deleted. line 59: Since no aerosols were measured, I wouldn't mention it in the abstract of the paper line 70: Quinn and Bates 2011 should be also cited since evidence for climate regulation though DMS still needs to be proven. line 92: misplacement of the ( line 149: the sentence should read "...the potential climatic relevant gas..." line 218: were the samples fixed with any fixative? P+G? line 221: Dinoflagellate abundance was determined? lines 314-325: Very interesting results that can be more discussed after Ruiz-Gonzalez et al. ISME Journal (2012) 6, 650–65, for instance. line 448: "Microbial affinity for DMSPd as indicated by" can be deleted line 651: I love Table 3 line 665: Could cyanobacteria be included? Were them measured by flow cytometry? It is a pity no taxonomical description of the communities could be performed. line 748: What about the role of algal oxidative stress? do you have any measurement indicating senescence of the bloom during the sampled period of time? line 789: "much needed" can be deleted.

---

## Referee Comment (RC2) · Anonymous Referee #2 · 23 Jun 2017

The manuscript reports on measurements of dimethyl sulfur compounds DMSC (DMS and DMSP) concentrations and their cycling rates on both sides of the Subtropical Front near New Zealand. The study is part of the SOAP experiment and intends to relate DMSPC dynamics to hydrographic and biological characteristics. To do so, measurements concentrate in three different areas that are investigated with a Lagrangian approach. The DMSP availability hypothesis is used as the major driver for the interpretation of most of the data, yet with uneven fit. The authors conclude that, as previously suggested, oceanic fronts generate hotspots for the production and emission of dimethyl

sulfur.

Even though no great advances in knowledge are provided that can be of applicability to a broad range of regions of the global ocean, the study is timely and the data valuable. The manuscript is well written and properly contextualized and referenced. I do not have major concerns towards publication but provide here below some questions and suggestions that may help improve the robustness and argumentation.

Methods, equation 1 and L206-213, also L541-550: SRD is calculated from daily-averaged irradiance. Is it taken for the 24 hours prior to sampling? Or is it the 24 hours of the sampling day? The rationale of the SRD concept related to DMS (as from Vallina & Simó 2007) relies on the previous 24 hours, which is the time over which photobiological and photochemical processes led to the observed DMS concentration.

L241-258: Provide details of how 35S-DMSPd loss was measured – I guess it was by removal of 35S-DMS, transformation of all the remaining 35S-DMSPd into 35S-DMS, which is trapped onto H2O2-soaked filter. Am I right?

L341-342: How was the cryogenic trap cooled to -20°C?

Results, L464-466: A bacterial DMS production rate (from DMSPd only) of 27 nmol/Ld is astonishingly high, more so when DMS concentration is 3 nmol/L and DMSPd is <10 nmol/L. It actually seems suspicious of mistake. I guess you have repeatedly checked up.

Discussion, L562: Cytosolic DMSP concentration should be given in fmol/um3 or similar (i.e., intracellular concentration) since pg/cell does not say much given the enormous size range of phytoplankton.

L616: The papers by Tortell et al. that emphasize DMS increases across oceanic fronts should be cited.

L643-669: To what extent the 50-fold range in DMSPd consumption rate constants cannot be due to methodological uncertainties in either DMSPd concentrations or the
35S experiments? This range factor seems very large, and the turnover at station 5 seems super fast (turnover time 1 h). More so when there is no correlation to bacterial abundance or production. I agree that bulk bacterial production holds less potential to drive DMSPd consumption than taxon-specific production, but a critical view of uncertainties is warranted. By the way, the range of turnover times shown in Table 3 for the present study is 0.1-1.6 d – if the fastest was 1 h, this should read 0.05-1.6.

L699-703: The relationship between the DMSPd-to-DMS conversion efficiency and rates of bacterial leucine incorporation is intriguing. You claim this is because as bacteria increase their C incorporation, they do it by cleaving more DMSP to use its C. I am not persuaded by the argument. Bacteria also increase their S demand, when increasing C incorporation. Why not taking up DMSPd as both a C and a S source? From the subsequent arguments, should we understand that abundance of other labile C forms (and potentially org S forms), bacteria exhibited low DMSP assimilation rates and rather they cleaved quite a share of the available DMSPd? But DMS yields were not particularly high either. Please clarify your arguments.

You could also invoke phycosphere-associated processes. In blooms like these there may be many bacteria closely associated to microalgae and therefore exposed to even higher concentrations of DMSP.

L778: Give range or std dev.

L775-787: To support the idea that phytoplankton-mediated DMS production largely contributed to gross DMS production, note that, in the DISCO experiment, Steinke et al. (AME 2002) found that the majority of potential DMSP-lyase activity occurred in particles >10 $\mu$m, namely dinoflagellates.

Figure 5: Correlation between DMSPt and chlorophyll a is quite strong indeed. One would expect it even stronger with DMSPp, since it is better associated with algal cells. Perhaps it does not deserve another graph but some mention to the regression facts.

Table 1: All variables are reported "in blooms" and in the vicinity (N or S of). But chlorophyll concentrations are not any lower in the vicinities. So, what is the definition of bloom? Same for nutrients and DMSP:Chla.

I like the data compilation in Table 3.

---

## Author Response (AR1)

1. The methodology used is correct and well described. As a general comment, the only weakness detected on this study is that not all pools of DMS(P) cycling were covered since no measurements of DMSO were performed (particulate and dissolved) which hampers a more extended discussion on the fate of metabolized DMS in seawater.

Answer ML. Indeed, DMSO measurements would have been a very appreciable addition to the paper. Unfortunately, they were not available, and we therefore cannot rule on the fate of certain pools.

2. It is really appreciated negative results of influence of light preincubations on DMSP dynamics. I think it is not stressed enough in the discussion of the paper. One thinks it is a pity than in such DMSP-active zone more specific experiments to test still open questions of the cycle, mainly related to the different physiological and ecological roles of DMSP in the upper ocean could have been tested (for instance, the relative role of non-DMSP-producers algae as sink of DMSP, algal DMS production, new in situ production of DMSP by heterotrophic bacteria, chemotaxis, etc). Rather than a weakness, I hope the paper will encourage the DMSP community to sample in the described area.

Answer ML. We agree with the reviewer. We furthered the discussion by adding some information in the methodology section (lines 334-348), please see point#11 of this review for the full description of the added information.

**3. line 38: there is more than only 2 fates of consumed DMSP, excretion as an oxidized form but not incorporated into cell structure is missed.**

Answer ML. To address this we changed the phrase: "This study focused on the two opposing fates of DMSP-S following its uptake by microbial organisms: either its conversion into DMS, or its assimilation into bacterial biomass." For the following: "This study focused **on two opposing short-term** fates of DMSP-S following its uptake by microbial organisms: either its conversion into DMS, or its assimilation into bacterial biomass."

Then we also added information about the third fate in the introduction section: "Another potential fate for DMSP is its transformation into dissolved non-volatile degradation products (DNVS), including sulfate (SO42-), however less is known of the molecular pathways involved in this process (Kiene et al. 2000; Reisch et al. 2011)."

*4. line 45. "measured in this study" can be deleted.* Answer ML. The words "…measured in this study…" have been deleted.

**5. line 59: Since no aerosols were measured, I wouldn't mention it in the abstract of the paper**

Answer ML. The following phrase: "The findings from this study provide crucial information on the distribution and cycling of DMS and DMSP in a critically undersampled area of the global ocean, and they highlight the importance of oceanic fronts as hotspots of the production of marine biogenic S compounds and as potential sources of aerosols particularly in regions of low anthropogenic perturbations such as the frontal waters of the Southern Hemisphere.", was changed to: "The findings from this study provide crucial information on the distribution and cycling of DMS and DMSP in a critically under-sampled area of the global ocean, and they highlight the importance of oceanic fronts as hotspots of the production of marine biogenic S compounds."

**6. line 70: Quinn and Bates 2011 should be also cited since evidence for climate regulation though DMS still needs to be proven.**

Answer ML. The following phrase: "DMS has gained notoriety over several decades of research on the grounds of its potential role linking ocean biology and the climate (Andreae et al., 1985; Charlson et al., 1987; Lovelock et al., 1972)." Was changed to: "DMS has gained notoriety over several decades of research on the grounds of its potential role linking ocean biology and the climate (Andreae et al., 1985; Charlson et al., 1987; Lovelock et al., 1972), a role that is still under debate (Quinn and Bates 2011, Quinn et al. 2017)."

**7. line 92: misplacement of the (**

Answer ML. The parenthesis in the following phrase conforms to the requirements: "These productive regions sometimes form unique biogeographic habitats of their own such as the Subtropical Convergence province proposed by Longhurst (2007)."

**8. line 149: the sentence should read "...the potential climatic relevant gas..."**

Answer ML. The following phrase: "Depending on bacterial requirements for either S or C and the relative contribution of DMSP to the overall oceanic S pool (Kiene et al. 2000; Levasseur et al 1996; Pinhassi et al. 2005), at least two very different and competing outcomes are involved from the bacterial catabolism of DMSP: one producing DMS, the climatic relevant gas, the other producing methanethiol (MeSH), an important microbial substrate (Kiene and Linn, 2000b).", was changed to: "Depending on bacterial requirements for either S or C and the relative contribution of DMSP to the overall oceanic S pool (Kiene et al. 2000; Levasseur et al 1996; Pinhassi et al. 2005), at least two very different and competing outcomes are involved from the bacterial catabolism of DMSP to the overall oceanic S pool (Kiene et al. 2000; Levasseur et al 1996; Pinhassi et al. 2005), at least two very different and competing outcomes are involved from the bacterial catabolism of DMSP to the overall oceanic S pool (Kiene et al. 2000; Levasseur et al 1996; Pinhassi et al. 2005), at least two very different and competing outcomes are involved from the bacterial catabolism of DMSP: one producing DMS, the **potential** climatic relevant gas, the other producing methanethiol (MeSH), an important microbial substrate (Kiene and Linn, 2000b).

**9. line 218: were the samples fixed with any fixative? P+G?**

Answer ML. No paraformaldehyde nor glutaraldehyde were used, rather the samples were snap-frozen in liquid N2 and quickly analyzed after that. The following phrase: "Bacterial samples were frozen in liquid nitrogen (Lebaron et al., 1998) and thawed immediately before counting by flow cytometry following the methods described in Safi et al. (2007)." Was changed to: Bacterial samples were **snap-frozen** in liquid nitrogen (Lebaron et al., 1998) and thawed immediately before counting by flow cytometry following the methods described in Safi et al. (2007)." Was changed to: Bacterial samples were **snap-frozen** in liquid nitrogen (Lebaron et al., 1998) and thawed immediately before counting by flow cytometry shortly after the cruise following the methods described in Safi et al. (2007).", in order to make it clearer.

**10. line 221: Dinoflagellate abundance was determined?**

Answer ML. Dinoflagellate abundance was determined in surface samples for all stations but not systematically for the "near surface" samples from which the incubation experiments were derived in this paper. It is thus possible to provide some information about the overall "regional" conditions of phytoplankton dominance shown in Table 1 but not to discuss the specific near surface abundances. The phytoplankton speciation data will be discussed in a separate DMS/marine biogeochemistry paper. Nevertheless we modified the information by adding a phrase: "Coccolithophore abundance **in near surface waters** was determined using optical microscopy as described in Chang and Northcote (2016)." **Dinoflagellate abundance was determined for surface waters (not for near surface waters) and is not shown here."**

**11. lines 314-325: Very interesting results that can be more discussed after Ruiz-Gonzalez et al. ISME Journal (2012) 6, 650–65, for instance.**

Answer ML. It is true that the absence of a significant difference between pre-incubation treatments is interesting in itself. We added some discussion on this, referring to Ruiz-González et al. (2012) and other publications (specifically related to the sulfur-relevant responses) but also more particularly to the review published by Ruiz-González et al in 2013 which clearly shows that the past 20 years of research on sunlight-bacteria interactions display a wide-range in responses (from negative to positive effect of natural sunlight on metabolic activity of heterotrophic bacteria) intimately linked with factors such as the phylogeny of bacterial groups under investigation, the light-history experienced by the natural populations, and many more. The added information is in bold in the following section:

"On the whole, the light conditions (dark and ambient) at which the cells were preacclimated for 6 h had no significant effect on the 35S-DMSPd metabolic rates measured. This result contrasts with findings from earlier studies (such as Galí et al., 2011; Ruiz-González et al., 2012a; Slezak et al., 2001, 2007; Toole et al., 2006) and could be related to a number of variables such as the timing and depth of sampling, the type of bacterial assemblages present and their previous light-history, as well as the different temporal and spatial scales at which exposure to solar radiation varies (Ruiz-González et al., 2013). Because of these wide-ranging and intricate lightbacteria interactions, natural solar radiation is believed to play a significant, yet challenging to predict, role in modulating bacterial dynamics and biogeochemical functions (Ruiz-González et al., 2013). In the current study, the sulfur-related metabolic activities of the marine biota sourced in the morning (between ca. 7h00 and 9h00; Table 1) from the highly irradiated near surface waters may have persisted in the dark within the time period of experimental pre-exposure (6 h), however the lack of information on the phylogeny of bacterial groups present, for example, hampers a more detailed discussion. We therefore present rate measurements made in dark-incubated samples that had been pre-exposed to ambient light conditions for 6 h."

**12. line 448: "Microbial affinity for DMSPd as indicated by" can be deleted**

Answer ML. Yes. The following phrase: "Microbial affinity for DMSPd, as indicated by the 35S-DMSPd loss rate constant ( $k_{DMSPd}$ ; Fig. 3a) varied between 0.4 and 3.4 d-1, with the exception of a higher value of 19.9 d-1 measured in the B2 cluster at station 5." Was

changed to: "The 35S-DMSPd loss rate constant ( $k_{DMSPd}$ ; Fig. 3a) varied between 0.4 and 3.4 d-1, with the exception of a higher value of 19.9 d-1 measured in the B2 cluster at station 5."

**13. line 651: I love Table 3**

**14. line 665: Could cyanobacteria be included? Were them measured by flow cytometry? It is a pity no taxonomical description of the communities could be performed.**

Answer ML. Yes it is indeed a good idea to mention cyanobacteria here as they have been shown (particularly Synechococcus and Prochlorococcus) to participate in DMSP assimilation. The following phrase: "It has been suggested that loss rate constants of DMSPd, rather than being directly related to stocks of bacteria could be more related to bacterial community composition, and particularly the specific abundance of Roseobacter, a member of Alphaproteobacteria, and with Gammaproteobacteria (Rover et al., 2010), which are both significant contributors to DMSP metabolism (Malmstrom et al., 2004a, 2004b; Vila-Costa et al., 2007; Vila et al., 2004).", was changed to: "It has been suggested that loss rate constants of DMSPd, rather than being directly related to stocks of bacteria could be more related to bacterial community composition, and particularly certain members of Alphaproteobacteria, Gammaproteobacteria as well as cyanobacteria, that could all potentially represent significant contributors to DMSP metabolism (Malmstrom et al., 2004a, 2004b, 2005; Royer et al., 2010; Vila-Costa et al., 2007; Vila et al., 2004). The appropriate references were also added (Vila-Costa et al 2006a as well as Malmstron et al. 2005). In reference to the other questions: we agree, it is highly unfortunate that no taxonomical description is available for the heterotrophic bacteria and picoplankton communities. This also limits our comprehension of the response of the biotic community under the different pre-incubation light exposure scenarios.

**15. line 748: What about the role of algal oxidative stress? do you have any measurement indicating senescence of the bloom during the sampled period of time?**

Answer ML: Measurements of photosynthetic efficiency (Fv/Fm) would have indeed been appreciable here, but are unfortunately not available. However we modified the phrase to reflect this possibility. The following phrase: "Community DMS production may have included indirect processes such as zooplankton grazing, viral lysis, and senescence, as well as direct algal DMSP-lyase activity associated with the presence of certain species of dinoflagellates and coccolithophores (Niki et al., 2000; Wolfe and Steinke, 1996), ubiquitous in Subantarctic waters in early March.", was changed to: "Community DMS production may have included indirect processes such as zooplankton grazing, viral lysis, and senescence, as well as direct algal DMSP-lyase activity associated with the presence of certain species of dinoflagellates and coccolithophores (Niki et al., 2000; Wolfe and Steinke, 1996), ubiquitous in Subantarctic waters in early March.", was changed to: "Community DMS production may have included indirect processes such as zooplankton grazing, viral lysis, and senescence, as well as direct algal DMSP-lyase activity associated with the presence of certain species of dinoflagellates and coccolithophores (Niki et al., 2000; Wolfe and Steinke, 1996), ubiquitous in Subantarctic waters in early March, and potential algal oxidative stress associated to light or nutrient availability (Stefels et al., 2007; Sunda et al., 2002).

16. line 789: "much needed" can be deleted.

Answer ML: The phrase "Our study provides much needed information on both concentrations and cycling of dimethylated sulfur compounds within waters of the New Zealand biogeochemical province (NEWZ) and more specifically in an oceanic frontal region." Was changed to: "Our study provides information on both concentrations and cycling of dimethylated sulfur compounds within waters of the New Zealand biogeochemical province (NEWZ) and more specifically in an oceanic frontal region."

Response Reviewer 2. os-2017-32 Lizotte et al.

The manuscript reports on measurements of dimethyl sulfur compounds DMSC (DMS and DMSP) concentrations and their cycling rates on both sides of the Subtropical Front near New Zealand. The study is part of the SOAP experiment and intends to relate DMSPC dynamics to hydrographic and biological characteristics. To do so, measurements concentrate in three different areas that are investigated with a Lagrangian approach. The DMSP availability hypothesis is used as the major driver for the interpretation of most of the data, yet with uneven fit. The authors conclude that, as previously suggested, oceanic fronts generate hotspots for the production and emission of dimethyl C1 sulfur. Even though no great advances in knowledge are provided that can be of applicability to a broad range of regions of the global ocean, the study is timely and the data valuable. The manuscript is well written and properly contextualized and referenced. I do not have major concerns towards publication but provide here below some questions and suggestions that may help improve the robustness and argumentation.

1. Methods, equation 1 and L206-213, also L541-550: SRD is calculated from dailyaveraged irradiance. Is it taken for the 24 hours prior to sampling? Or is it the 24 hours of the sampling day? The rationale of the SRD concept related to DMS (as from Vallina & Simó 2007) relies on the previous 24 hours, which is the time over which photobiological and photochemical processes led to the observed DMS concentration.

Answer ML. The calculations are indeed based on the daily irradiance averaged over the 24 hours prior to sampling. We made this clearer in the methodology by modifying the following sentence: "Solar radiation dose (SRD in W m-2) was calculated using Eq. (1) where  $I_0$  represents the daily-averaged irradiance (in W m-2) measured using an Eppley Precision Spectral Pyronometer (285-2800 nm), k (in m-1) are estimates of vertical diffuse attenuation coefficients based on Photosynthetically Active Radiation (PAR) offset between two depths (2 m and 10 m), MLD is the mixed layer depth defined as the point at which a 0.2°C difference from the sea surface temperature occurred and was calculated according to Kara et al. (2000).", and changing it to: "Solar radiation dose (SRD in W m-2) was calculated using Eq. (1) where  $I_0$  represents the daily-averaged irradiance of the 24 hours prior to sampling (in W m-2) measured using an Eppley Precision Spectral Pyronometer (285-2800 nm), k (in m-1) are estimates of vertical diffuse attenuation coefficients based on Photosynthetically Active Radiation (PAR) offset between two depths (2 m and 10 m), MLD is the mixed layer depth defined as the point at which a 0.2°C difference from the sea surface temperature occurred and was calculated according to Kara et al. (2000).

2. L241-258: Provide details of how 35S-DMSPd loss was measured – I guess it was by removal of 35S-DMS, transformation of all the remaining 35S-DMSPd into 35S-DMS, which is trapped onto H2O2-soaked filter. Am I right?

Answer ML. The 35S-DMSPd loss rate is measured by the disappearance of dissolved 35S-DMSP over time: the loss of 35S-DMSPd reflecting what is being consumed. To add clarity to this part of the paper we included more information by modifying the following sentences: "The bottles were then incubated for 3 h at *in situ* temperature during which time subsamples were taken after 0, 30, 60, and 180 min to measure the loss of 35S-DMSPd over time. The kDMSPd was calculated as the slope of the natural log of the fraction of remaining 35S-DMSPd versus time." to these ones: "The bottles were taken after 0, 30, 60, and 180 min time **1mL** subsamples were taken after 0, 30, 60, and 180 min and transferred into 10-mL scintillation vials containing **5 mL Ecolume**TM in order to measure the loss of 35S-DMSPd over time (the disappearance of 35S-DMSPd representing the consumption of this pool). The kDMSPd was then calculated as the slope of the natural log of the natural log of the natural log of the natural log of the sentence."

The "transformation of all the remaining 35S-DMSPd into 35S-DMS, which is trapped onto H2O2-soaked filter" mentioned by the reviewer is called the "unreacted or unconsumed dissolved 35S-DMSP" which was measured at the end of the incubation period. We discuss this around lines 279-283: "After the volatiles were trapped, a new stopper with H2O2-soaked filter was placed in the vial. Each vial was then injected with 0.2 mL NaOH (5N) through the stopper using a BD precision guide needle to quantitatively cleave remaining 35S-DMSPd into 35S-DMS. The 35S-DMS was trapped as described above." To make it clearer what this pool represents we modified the phrase which now reads as follows: "After the volatiles were trapped, a new stopper with H2O2soaked filter was placed in the vial. Each vial was then injected with 0.2 mL NaOH (5N) through the stopper using a BD precision guide needle to quantitatively cleave the remaining 35S-DMSPd into 35S-DMS (**a pool known as the unconsumed 35S-DMSPd**). The 35S-DMS was trapped as described above."

**3. L341-342: How was the cryogenic trap cooled to -20°C?**

Answer ML. The trap was encased in a metal block that also contained a cold finger connected to an external cryo-cooling unit monitored and controlled automatically. The following phrase: "Briefly, calibrated volumes (5 mL) of seawater samples were purged with zero-grade nitrogen (99.9 % pure) and gas-phase DMS was cryogenically concentrated on 60/80 Tenax TA in a stainless steel trap at -20°C, then thermally desorbed at 100 °C for analysis by GC coupled with sulfur chemiluminescent detection.", was changed to: "Briefly, calibrated volumes (5 mL) of seawater samples were purged with zero-grade nitrogen (99.9 % pure) and gas-phase DMS was cryogenically concentrated on 60/80 Tenax TA in a stainless steel trap at -20°C, then thermally desorbed at 100 °C for analysis by GC coupled with sulfur chemiluminescent detection.", was changed to: "Briefly, calibrated volumes (5 mL) of seawater samples were purged with zero-grade nitrogen (99.9 % pure) and gas-phase DMS was cryogenically concentrated on 60/80 Tenax TA in a stainless steel trap **maintained at -20°C via a cold finger connected to a cryo-cooling unit**, then thermally desorbed at 100 °C for analysis by GC coupled with sulfur chemiluminescent detection."

4. Results, L464-466: A bacterial DMS production rate (from DMSPd only) of 27 nmol/Ld is astonishingly high, more so when DMS concentration is 3 nmol/L and DMSPd is 10  $\mu$ m, namely dinoflagellates.

Answer ML. We agree with the reviewer, this rate is quite high. Such high rates are rare but have been published before (Royer et al. 2010). We are confident however that this is not a problem with a specific incubation (or bottle effect) since all the incubations displayed the same results (even the duplicate dark-acclimated samples that we do not present in the paper, as mentioned in the methodology section, were extremely high and showed no significant differences with the light-acclimated samples that we discuss in the paper). We added the following phrases to this section in order to reflect potential reasons for this response: "This high rate reflects the very high DMSPd scavenging by the bacteria measured on this particular day. The fact that concentrations of DMS remained low (ca. 3 nmol L-1) suggests that potential sinks, particularly bacterial DMS consumption, but not excluding DMS photo-oxidation and ventilation (Table 1) may have kept this pool in check."

5. Figure 5: Correlation between DMSPt and chlorophyll a is quite strong indeed. One would expect it even stronger with DMSPp, since it is better associated with algal cells. Perhaps it does not deserve another graph but some mention to the regression facts.

Answer ML. The strength of the regression between DMSPp and Chl a (r2 = 0.57) is very similar to the one between DMSPt and Chl a (r2 = 0.59). We added this information in the discussion section (starting at line 541): "A type II linear regression model suggests that 59% of the variance in pools of DMSPt can be explained by the variability in stocks of chl a (Fig. 5a), while the strength of the relationship between DMSPp and chl a is also strong (r2 = 0.57, data not shown)."

6. C3 Table 1: All variables are reported "in blooms" and in the vicinity (N or S of). But chlorophyll concentrations are not any lower in the vicinities. So, what is the definition of bloom?

Answer ML. This is discussed in detail by the SOAP overview paper (Law et al. also currently under review). However to add some precision to this aspect we added the following phrase to the methodology section (lines 204-209). "The SOAP blooms were coherent discrete areas of elevated ocean colour identified in satellite images characterised by a maximum of  $1 \text{ mg/m}^3$  chl *a* or higher. Sampling took place near the center of these blooms but also at stations on the periphery and outside the blooms (Table 1), as defined by the distance from the bloom centre and clear demarcation in surface biogeochemical variables (see Law et al., this issue)."

In this paper, we separate the stations in "clusters" (see all figures and discussions ensuing), to account for the fact that stations are either directly "in" or "in the vicinity of" the blooms.

**7. Same for nutrients and DMSP: Chla.**

Answer ML. We are not certain what the reviewer is asking here. If possible, added information would help us address any concerns regarding this part of the paper. Thank you.

8. *I like the data compilation in Table 3.* Answer ML. We thank both reviewers for acknowledging this positive aspect of the paper.

---

## Editor Decision (ED1)

15 Sep 2017

Thanks for the detailed responses and track changed modifications in response to reviewers RC-1 and RC-2.

**Further responses required:**

Please can you also respond to RC-2 points that you have not responded to.

*L616: The papers by Tortell et al. that emphasize DMS increases across oceanic fronts should be cited.*

*L643-669: To what extent the 50-fold range in DMSPd consumption rate constants cannot be due to methodological uncertainties in either DMSPd concentrations or the 35S experiments? This range factor seems very large, and the turnover at station 5 seems super fast (turnover time 1 h). More so when there is no correlation to bacterial abundance or production. I agree that bulk bacterial production holds less potential to drive DMSPd consumption than taxon-specific production, but a critical view of uncertainties is warranted. By the way, the range of turnover times shown in Table 3 for the present study is 0.1-1.6 d – if the fastest was 1 h, this should read 0.05-1.6.*

*L699-703: The relationship between the DMSPd-to-DMS conversion efficiency and rates of bacterial leucine incorporation is intriguing. You claim this is because as bacteria increase their C incorporation, they do it by cleaving more DMSP to use its C. I am not persuaded by the argument. Bacteria also increase their S demand, when increasing C incorporation. Why not taking up DMSPd as both a C and a S source? From the subsequent arguments, should we understand that abundance of other labile C forms (and potentially org S forms), bacteria exhibited low DMSP assimilation rates and rather they cleaved quite a share of the available DMSPd? But DMS yields were not particularly high either. Please clarify your arguments. You could also invoke phycosphere-associated processes. In blooms like these there may be many bacteria closely associated to microalgae and therefore exposed to even higher concentrations of DMSP.*

*L778: Give range or std dev.*

*L775-787: To support the idea that phytoplankton-mediated DMS production largely contributed to gross DMS production, note that, in the DISCO experiment, Steinke et al. (AME 2002) found that the majority of potential DMSP-lyase activity occurred in particles >10 m, namely dinoflagellates.*

**RC-1, point 3**

– suggest it is useful to reiterate here:

Following "assimilation into bacterial biomass"
with
"and has not considered dissolved non-volatile degradation products."

The addition:
"Dinoflagellate abundance was determined for surface waters (not for near surface waters) and is not shown here."
is not particularly useful to the reader.  Can a reference to data be given or numbers included in Table 1?

**RC-2 point 5**

Suggest reword:
"while the strength of the relationship between DMSPp and chl a is also strong (r2 = 0.57, data not shown)."

With
"while the correlation between DMSPp and chl a is of similar strength (r2 = 0.57, data not shown)."

**RC-2 point 6**

With the addition:
"The SOAP blooms were coherent discrete areas of elevated ocean colour identified in satellite images characterised by a maximum of 1 mg/m3 chl a or higher. Sampling took place near the center of these blooms but also at stations on the periphery and outside the blooms (Table 1), as defined by the distance from the bloom centre and clear demarcation in surface biogeochemical variables (see Law et al., this issue)."

I believe this should read:
"The SOAP blooms were coherent discrete areas of elevated ocean colour identified in satellite images characterised by a minimum of 1 mg m$^{-3}$ chl a or higher. Sampling took place near the center of these blooms but also at stations on the periphery and outside the blooms (Table 1), as defined by the distance from the bloom centre and clear demarcation in surface biogeochemical variables (see Law et al., this issue)."

Saying blooms are chl-a areas up to 1 mg m$^{-3}$ or greater sets no limits at all!  I think this should read "by a minimum" rather than "by a maximum"

**RC-2 point 7 (and parts of 6)**

You say: "We are not certain what the reviewer is asking here. If possible, added information would help us address any concerns regarding this part of the paper."

I read that the reviewer is questioning the partitioning of sample sites between "in" the bloom and "in the vicinity" of the bloom and you do mention that this is a geographic distinction -   Would it be more accurate to replace

"and clear demarcation in surface biogeochemical variables (see Law et al..."
with:

"determined from pre-site surveys with bloom centre marked by drifting spar buoy (see Law et al...."""

I read that the reviewer questions variables in Table 1 including Chl-a, nutrients and DMSP:Chla that do not show clear differences related to e.g. nutrient drawdown in bloom or greatly elevated Chla or DMSP in the bloom compared with the 2 stations north and south of blooms. (Perhaps this can be addressed by discussing that stations adjacent to bloom were also in generally productive waters).

**Additional corrections:**

I note error in footnote to Table 1
Change "then the 9 presented"
to "than the 9 presented"

---

## Author Response (AR2)

Thanks for the detailed responses and track changed modifications in response to reviewers RC-1 and RC-2.

Further responses required: Please can you also respond to RC-2 points that you have not responded to.

**Answer ML.** I sincerely apologize. I'm not sure how these points were lost from the review. In any case, please accept my apologies.

L616: The papers by Tortell et al. that emphasize DMS increases across oceanic fronts should be cited.

**Answer ML.** Absolutely, they should. The following references from the Tortell group were added to this part of the discussion as well as in the list of citations: Asher et al. 2017, Nemcek et al. 2008, Tortell et al. 2005, Tortell and Long 2009. "The heightened biological activity in these regions (Llido et al., 2005) is thought to lead to intensified carbon drawdown on seasonal timescales (Metzl et al., 1999) as well as high concentrations of DMS (Asher et al., 2017; Holligan et al., 1987; Matrai et al., 1996; Nemcek et al., 2008; Tortell, 2005; Tortell and Long, 2009)."

L643-669: To what extent the 50-fold range in DMSPd consumption rate constants cannot be due to methodological uncertainties in either DMSPd concentrations or the 35S experiments? This range factor seems very large, and the turnover at station 5 seems super fast (turnover time 1 h). More so when there is no correlation to bacterial abundance or production. I agree that bulk bacterial production holds less potential to drive DMSPd consumption than taxon-specific production, but a critical view of uncertainties is warranted. By the way, the range of turnover times shown in Table 3 for the present study is 0.1-1.6 d – if the fastest was 1 h, this should read 0.05-1.6.

**Answer ML**. We understand the concerns put forward by the reviewer here. We certainly cannot exclude any methodological uncertainties and analytical limitations, as is the case in any type of experimental setup. In order to address this we further discuss the potential caveats associated with methodology (measurements of $K_{DMSPd}$ and $DMSP_d$) and we calculate a factor of error propagation as the estimation of $DMSP_d$ consumption rates by multiplication of the $DMSP_d$ loss rate constants ($k_{DMSPd}$) with *in situ* $DMSP_d$ concentration carries a larger uncertainty. The error propagation was calculated by adding the relative uncertainties in quadrature (square root of the sum of squares).
Here are the changes (in bold) made to this part of the methodology (lines 327 and beyond):

"The measurement of the above variables allowed us to estimate $DMSP_d$ loss rate constants ($k_{DMSPd}$), **DMSPd turnover rates (or consumption rates) by multiplying values of $k_{DMSPd}$ with *in situ* $DMSP_d$ concentration, and** rates of gross DMS

production from DMSP$_d$ by multiplying values of **DMSP$_d$ turnover rates with** DMS yields. **We calculated the propagation of uncertainty for rates that represent estimations based on other measured variables by adding the relative error of each variable in quadrature and expressing them as percentages. The uncertainty associated with estimates of DMSP$_d$ turnover rates and DMS production rates from DMSP$_d$ were on average 35% and 37%, respectively. Furthermore, we cannot rule out any bottle effects during incubation experiment, nor can we dismiss potential filtration artefacts related to the determination of DMSP$_d$ concentrations with which the derived estimates are based on. However all measurements were made following the best practices published and available at the time of sampling. Finally**, the microbial transformation rates of DMSP$_d$ measured during these incubations are considered to stem mostly from bacterial processes however phytoplankton-related processes cannot be totally excluded as low DMSP-producing phytoplankton and picophytoplankton have been shown to assimilate DMSP$_d$-sulfur (Malmstrom et al., 2005; Ruiz-González et al., 2011; Vila-Costa et al., 2006b).

Concerning the last part of the comment referring to values in Table 3: The values were rounded to 1 digit of precision for the purposes of uniformity in Table 3, thus transforming 0.05d to 0.1d as the lower limit within the range for this study. We agree that this makes a difference so the range of values presented in Table 3 were kept at 2 digits of precision to reflect the precise turnover time.

L699-703: The relationship between the DMSPd-to-DMS conversion efficiency and rates of bacterial leucine incorporation is intriguing. You claim this is because as bacteria increase their C incorporation, they do it by cleaving more DMSP to use its C. I am not persuaded by the argument. Bacteria also increase their S demand, when increasing C incorporation. Why not taking up DMSPd as both a C and a S source? From the subsequent arguments, should we understand that abundance of other labile C forms (and potentially org S forms), bacteria exhibited low DMSP assimilation rates and rather they cleaved quite a share of the available DMSPd? But DMS yields were not particularly high either. Please clarify your arguments. You could also invoke phycosphere-associated processes. In blooms like these there may be many bacteria closely associated to microalgae and therefore exposed to even higher concentrations of DMSP.

**Answer ML.** We provide further details in order to clarify our arguments by adding these lines (**in bold**) to section 5.4.3:

[revised manuscript text omitted]

L778: Give range or std dev.

**Answer ML.** We added the std deviations in this part of the discussion and added the word "mean": "Coarse calculations that assume steady-state conditions suggest that transposing these net changes over a daily period amounts to a mean net community production of DMS from $DMSP_t$ of **15.2 ± 16.4 nmol L$^{-1}$ d$^{-1}$** (n = 6) throughout the stations where data was available. This rough **mean** estimate is almost 3 times as high as the gross microbial production of DMS from $DMSP_d$ (average of **5.3 ± 9.9 nmol L$^{-1}$ d$^{-1}$**, n = 6) in the same stations (sta. 3, 5, 6, 7, 8 and 9)."

L775-787: To support the idea that phytoplankton-mediated DMS production largely contributed to gross DMS production, note that, in the DISCO experiment, Steinke et al. (AME 2002) found that the majority of potential DMSP-lyase activity occurred in particles >10 m, namely dinoflagellates.

**Answer ML.** Although estimates of the potential lyase activity are difficult to transpose to the natural environment (because these rates are measured on extracted enzymes at saturating DMSP concentrations) we added the following phrase (and reference), in bold, to this section of the discussion: "The microbial DMS production rates from $DMSP_d$ in this study are also considerably lower than several of the community net production rates required to support microlayer DMS (range of -1445 to 5529 nmol L$^{-1}$ h$^{-1}$) reported by Walker et al. (2016). **Estimates of the relative importance of phytoplankton-mediated DMS production are scarce, however a study conducted in waters of the North Atlantic during a summer coccolithophore bloom suggested that as much as 74% of the potential DMSP-lyase activity occurred in the > 10 µm particulate fraction, which contained a high proportion of dinoflagellates (Steinke et al., 2002).** Altogether our findings support the view that indirect and direct processes of phytoplankton-mediated DMS production were important contributors to standing stocks of DMS in the near-surface waters of the STF during austral summer."

RC-1, point 3 – suggest it is useful to reiterate here: Following "assimilation into bacterial biomass" with "and has not considered dissolved non-volatile degradation products."

**Answer ML.** We modified the phrase and added the following words (in bold): This study focused on two opposing short-term fates of DMSP-S following its uptake by microbial organisms: either its conversion into DMS, or its assimilation into bacterial biomass, **and has not considered dissolved non-volatile degradation products.**

RC-1, point 10 - The addition: "Dinoflagellate abundance was determined for surface waters (not for near surface waters) and is not shown here." is not particularly useful to the reader. Can a reference to data be given or numbers included in Table 1?

Answer ML. The phytoplankton data (including abundance and carbon content of dinoflagellates and other groups) will be addressed in a separate paper that is yet to be submitted. The following phrase was deleted: "Dinoflagellate abundance was determined for surface waters (not for near surface waters) and is not shown here.", and was changed (at lines 242-245) for "**No further information regarding the abundance of eukaryotic organisms in near surface waters is available however the abundance and carbon content of other groups of phytoplankton in surface waters will be discussed in a separate paper relating DMS cycling and marine biogeochemistry (*C. Law, personal comm.*).**"

RC-2 point 5 Suggest reword: "while the strength of the relationship between DMSPp and chl a is also strong (r2 = 0.57, data not shown)." With "while the correlation between DMSPp and chl a is of similar strength (r2 = 0.57, data not shown)."

**Answer ML.** We changed the wording of the phrase (at line 565), which now reads as follows: "A type II linear regression model suggests that 59% of the variance in pools of $DMSP_t$ can be explained by the variability in stocks of chl *a* (Fig. 5a), **while the correlation between $DMSP_p$ and chl *a* is of similar strength ($r^2$ = 0.57, data not shown).**"

RC-2 point 6 With the addition: "The SOAP blooms were coherent discrete areas of elevated ocean colour identified in satellite images characterised by a maximum of 1 mg/m3 chl a or higher. Sampling took place near the center of these blooms but also at stations on the periphery and outside the blooms (Table 1), as defined by the distance from the bloom centre and clear demarcation in surface biogeochemical variables (see Law et al., this issue)." I believe this should read: "The SOAP blooms were coherent discrete areas of elevated ocean colour identified in satellite images characterised by a minimum of 1 mg m-3 chl a or higher. Sampling took place near the center of these blooms but also at stations on the periphery and outside the blooms (Table 1), as defined by the distance from the bloom centre and clear demarcation in surface biogeochemical variables (see Law et al., this issue)." Saying blooms are chl-a areas up to 1 mg m-3 or greater sets no limits at all! I think this should read "by a minimum" rather than "by a maximum"

**Answer ML.** Yes absolutely, we changed the wording (at lines 217-222) as recommended (also including part of the next comment RC-2 point 7 just below): "The SOAP blooms were coherent discrete areas of elevated ocean colour identified in satellite images characterised by a minimum of 1 mg m$^{-3}$ chl *a* or higher. Sampling took place near the center of these blooms but also at stations on the periphery and outside the blooms (Table 1), as defined by the distance from the bloom centre determined from pre-site surveys with bloom centre marked by a drifting spar buoy (see Law et al., this issue)."

RC-2 point 7 (and parts of 6) You say: "We are not certain what the reviewer is asking here. If possible, added information would help us address any concerns regarding this

part of the paper." I read that the reviewer is questioning the partitioning of sample sites between "in" the bloom and "in the vicinity" of the bloom and you do mention that this is a geographic distinction - Would it be more accurate to replace "and clear demarcation in surface biogeochemical variables (see Law et al..." with: "determined from pre-site surveys with bloom centre marked by drifting spar buoy (see Law et al....""" I read that the reviewer questions variables in Table 1 including Chl-a, nutrients and DMSP:Chla that do not show clear differences related to e.g. nutrient drawdown in bloom or greatly elevated Chla or DMSP in the bloom compared with the 2 stations north and south of blooms. (Perhaps this can be addressed by discussing that stations adjacent to bloom were also in generally productive waters).

**Answer ML.** Thank you very much for the precisions. As mentioned above (comment RC-2 point 6) we first changed the following phrase to complete information on the sampling strategy: "The SOAP blooms were coherent discrete areas of elevated ocean colour identified in satellite images characterised by a minimum of 1 mg m$^{-3}$ chl *a* or higher. Sampling took place near the center of these blooms but also at stations on the periphery and outside the blooms (Table 1), as defined by the distance from the bloom centre **determined from pre-site surveys with bloom centre marked by a drifting spar buoy** (see Law et al., this issue)." We also added the following phrase to the methods section (line 222 and beyond): "**Note that stations adjacent to the blooms were also located in generally productive waters (Table 1)**."

Additional corrections: I note error in footnote to Table 1 Change "then the 9 presented" to "than the 9 presented
**Answer ML.** Done, the word "then" was changed to "than"

---

## Author Response (AR3)

Editor feedback to author response: os-2017-32 Lizotte et al. 16 Oct 2017

Editor
Line801:
(see Review by Ramanan et al., 2016)
change to
(see review by Ramanan et al., 2016)

Response ML:

Line 801 (see Review by Ramanan et al., 2016)
was changed to
(see review by Ramanan et al., 2016)

I have accepted all the changes made to the last version of the revised document (including this last change "Review" for "review") and will upload this version for final submission. If need be, the marked-up version can also be sent.

As stated in the author instructions, the text must include Tables embedded directly in the document (not as pictures). "*Please make sure that your text is complete and includes the title, authors and their affiliations, abstract, tables (not included as images), and figure and table captions*". Tables 1 and 2 have been inserted as word Tables however Table 3 is too large (it exceeds the maximum of 10 columns possible). For now it is included as an image (pdf), I will await further instruction concerning this specific Table.